# A Comprehensive Benchmark for RNA 3D Structure–Function Modeling

## Abstract

The relationship between RNA structure and function has recently attracted interest within the deep learning community, a trend expected to intensify as nucleic acid structure models advance. Despite this momentum, the lack of standardized, accessible benchmarks for applying deep learning to RNA 3D structures hinders progress. To this end, we introduce a collection of seven benchmarking datasets specifically designed to support RNA structure–function prediction. Built on top of the established Python package `rnaglib`, our library streamlines data distribution and encoding, provides tools for dataset splitting and evaluation, and offers a comprehensive, user-friendly environment for model comparison. The modular and reproducible design of our datasets encourages community contributions and enables rapid customization. To demonstrate the utility of our benchmarks, we report baseline results for all tasks using a relational graph neural network. Data and code at `anonymous.4open.science/r/rnaglib-7596/`[1]

## 1 Introduction

Recent years have witnessed the advent of deep learning methods for structural biology, culminating in the award of the Nobel Prize in Chemistry in 2024. AlphaFold (Jumper et al., 2021) revolutionized protein structure prediction, equipping the field with millions of new structures. Breakthroughs go beyond structure prediction, notably in protein design (Watson et al., 2023; Dauparas et al., 2022), drug discovery (Schneuing et al., 2024; Corso et al., 2022) and fundamental biology (van Kempen et al., 2022). A key factor in these breakthroughs was the development of neural encoders that directly model protein structure (Jing et al., 2020; Zhang et al., 2022b; Gainza et al., 2020; Wang et al., 2022). These advances in turn are rooted in robust competitions (CASP, CAPRI) and benchmarks (Townshend et al., 2021a; Kucera et al., 2023; Zhu et al., 2022; Jamasb et al., 2024; Notin et al., 2023). By setting clear goals, such benchmarks are the foundation for the development of structure encoders and are required to leverage ever-increasing structural data availability. Yet, to date, structure–function benchmarks have focused on proteins, hampering model development for other equally important biological macromolecules.

Ribonucleic acids (RNAs) are a large family among these macromolecules, supporting biological functions along every branch in the tree of life. Besides messenger RNAs which code for proteins, non-coding RNAs are a vast class of molecules which can adopt complex 3D folds (Cech and Steitz, 2014) and play diverse roles in cellular functions, including gene regulation, RNA processing, and protein synthesis (Statello et al., 2021). However, our understanding of non-coding RNAs and their functions remains limited. This can be largely attributed to the negatively charged nature of RNA backbones, which makes them flexible and limits the availability of high-resolution RNA structures, and poses significant modeling challenges. Another predominant challenge to a functional understanding of RNA 3D structure lies in the lack of infrastructure for the development and evaluation of function prediction models. In this work, we propose a benchmarking suite to serve as a facilitating library for the development of structure encoders on RNA.

Our key contributions are:
- *Seven tasks related to RNA 3D structure* that represent various biological challenges. Each task consists of a dataset, a splitting strategy, and an evaluation method, laying the ground for comparable, reproducible model development.

---

[1]Note to reviewers: identifiable information refers to authors of the base package, not our additional library.

- *Modular annotators, filters and splitting strategies*, both novel and from existing literature, that facilitate the construction of new and custom tasks by other researchers, reducing the time spent on data preparation while ensuring reproducibility of achieved results.
- *We apply the benchmark* to simple architectures to initialize a leaderboard and investigate the effect of splitting, structural context, and RNA representation choice on performance.

## 2 RELATED WORK

### 2.1 PROTEIN FOCUSED BENCHMARKING

Classic tasks on proteins appeared independently in unrelated papers, such as GO term and EC number prediction (Gligorijević et al., 2021), fold classification (Hou et al., 2018), binding site detection and classification (Gainza et al., 2020) or binding affinity regression (Wang et al., 2005). *ATOM3D* (Townshend et al., 2021a) was the first systematic benchmark for molecular systems, albeit heavily focused on proteins. More comprehensive tools were proposed, such as *ProteinShake* (Kucera et al., 2023) *ProteinWorkshop* (Jamasb et al., 2024) and *TorchDrug* (Zhu et al., 2022), that unify the above tasks and lower the barrier to develop protein structure encoders. *ProteinGym* (Notin et al., 2023) addresses the evaluation of protein mutation effects, while Buttenschoen et al. (2024); Kovtun et al. (2024); Durairaj et al. (2024) address protein interactions. These works are notable efforts to scale datasets using predicted structures and to propose biologically appropriate splitting strategies.

### 2.2 RNA STRUCTURAL DATASETS AND BENCHMARKING

In RNA 3D structure-based modeling infrastructure, three papers propose machine learning ready datasets. They either provide cleaned files in various formats through a web interface (*RNAsolo* (Adamczyk et al., 2022)), join RNA structures with their sequence alignment (*RNANet* (Becquey et al., 2021)) or focus on structure prediction (*RNA3DB* (Szikszai et al., 2024)). Recently, multiple benchmarking suites were introduced for models on RNA sequences (*BEACON* (Ren et al., 2024)), predicted structures of short RNAs (Xu et al., 2024), or for fitness prediction (Arora et al., 2025). None of these methods propose benchmark tasks to compare RNA modeling and learning methods on experimental structures. Our effort is integrated within a previously published python package, `rnaglib` (Mallet et al., 2022) that facilitates the use of RNA structural data with 2.5D graphs. This work adds all necessary material to process data and datasets, as well as a set of seven tasks (see Supplementary Section A for a more detailed comparison).

### 2.3 STRUCTURE-BASED RNA MODELS

Whilst most deep learning models on RNA focus on secondary structure prediction and sequence-level tasks, some structure-based models were developed. *RBind* (Wang et al., 2018) proposed learning on RNA structures using residue graph representations, followed by others integrating sequence features in the learning (Su et al., 2021; Wang et al., 2023). *RNAmigos* (Oliver et al., 2020) proposed incorporating non-canonical interactions in the graph, in conjunction with metric learning pretraining. *ARES* (Townshend et al., 2021b) applied a tensor field neural network (Thomas et al., 2018) to RNA structure, modeling it as an atomic graph. *Geometric Vector Perceptron* (Jing et al., 2021), an equivariant message passing protein encoder, was adapted to RNA in (Joshi et al., 2024; Tan et al., 2023; 2025; Huang et al., 2024; Wong et al., 2024). *EquiRNA* (Li et al., 2025) proposed a hierarchical architecture. Recently, *HARMONY* (Xu et al., 2025) introduces a model using multiple RNA modalities.

Despite booming interest in deep learning-based modeling RNA 3D structure, we lack the means to systematically compare models and onboard new practitioners.

## 3 TOOLS TO ASSEMBLE TASKS

We describe the full process of building a task: data collection, quality filtering, cleanup, and splitting. Tasks are ready to use and documented for any practitioner wishing to build their own reproducible tasks with minimal effort.

**Data collection and annotation** Our data originates from the RCSB-PDB DataBank (Kouranov et al., 2006), where we fetch all 3D structures containing RNA. In some tasks, we use the non-redundant structures subset proposed in (Leontis and Zirbel, 2012) and later referred to as `bgsu`. We annotate each system with RNA-level features such as its resolution, and residue-level features such as coordinates, presence of interacting compounds or number of protein atoms in the residues' vicinities.

**Structure partitioning and quality filters** RNA molecules present in PDB files exhibit a bimodal distribution over the number of residues (see Supplementary Figures 5a and 5b). Many systems have less than 300 residues, while some (mostly rRNAs) have several thousands. We treat RNA fragments from the same PDB file as multiple systems if they do not interact, decreasing system size and reducing required computing power for downstream steps.

We implement resolution and size filters, with default cutoffs including systems below 4Å resolution and between 15 to 500 residues; the upper size cutoff limits computational expense. A novel protein content filter removes RNA structures heavily dependent on protein interactions, a crucial step overlooked in most existing datasets. We also include the drug-like filter from *Hariboss* (Panei et al., 2022) to screen small molecules binding to RNA.

**Redundancy filtering and dataset splitting** Rigorous data splitting is often key for structural biology benchmarks such as the *PINDER* database (Kovtun et al., 2024) for proteins, which has set a standard in the field for model generalization assessment. Given a set of RNA molecules $\mathcal{R}$, a similarity metric $s$ and a threshold $\theta$, let $\mathcal{G}$ be the graph connecting similar RNAs, $\mathcal{G} = (\mathcal{R}, \mathcal{E}^\theta)$, where $\mathcal{E}^\theta := \{(r_1, r_2) \in \mathcal{R}^2, s(r_1, r_2) > \theta\}$. We propose a clustering algorithm that returns connected components of $\mathcal{G}$ as clusters, ensuring that points in different clusters have a maximal similarity of $\theta$. We compute similarity $s$ with *CD-HIT* (Fu et al., 2012) for sequence and with *US-align* (Zhang et al., 2022a) for structure.

Building on this, we propose a redundancy removal step by selecting the highest-resolution RNA in each cluster. For most tasks, we use a 0.9 sequence similarity threshold and then 0.8 for structural similarity. We also propose a data splitting algorithm that aggregates clusters. To prevent data leakage, we build dissimilar clusters with a conservative structural similarity threshold (0.5 unless mentioned otherwise). Then, we cast the cluster aggregation into splits of given sizes as an integer programming problem, solved with the linear programming tool *PuLP* (Mitchell et al., 2011); this enforces structural dissimilarity and allows using label stratification.

**Metrics computation** Given the type of prediction target (e.g., binary classification, multilabel, etc.), each task implements a standardized way to compute evaluation metrics. Each task is evaluated by a recommended metric, specified in Table 1, and a range of additional standard metrics.

**Anatomy of a task** A "task" is now simply the collection of the aforementioned components with the choices relevant to a particular biological problem. That is, each task consists of (i) a dataset drawn from our 3D structure database and processed using our annotations, partitions and filters, (ii) redundancy removing and fixed splits based on sequence, structure or existing literature, and (iii) a well-defined evaluation protocol. This modular design will help other practitioners propose additional benchmark tasks, and lower the barrier to entry for training models on RNA structure.

## 4    TASKS ON RNA 3D STRUCTURE

Our tasks suite explores various dimensions of RNA structural research illustrated and briefly summarized in Figure 1. We offer seven tasks among which three are based on previous research with published datasets (RNA-IF (Joshi et al., 2024), RNA-Site (Su et al., 2021) and RNA-VS (Carvajal-Patiño et al., 2025)). For RNA-IF and RNA-Site, we propose an enhanced, expanded version, resulting in a total of *nine separate datasets*.

Armed with an RNA structure, the focus of each of the tasks is to predict aspects of the RNA's biological function, or to reverse the direction and predict the sequence from which a given structure arises (RNA design). Further information on task datasets is found in Table 1 and supplement C.

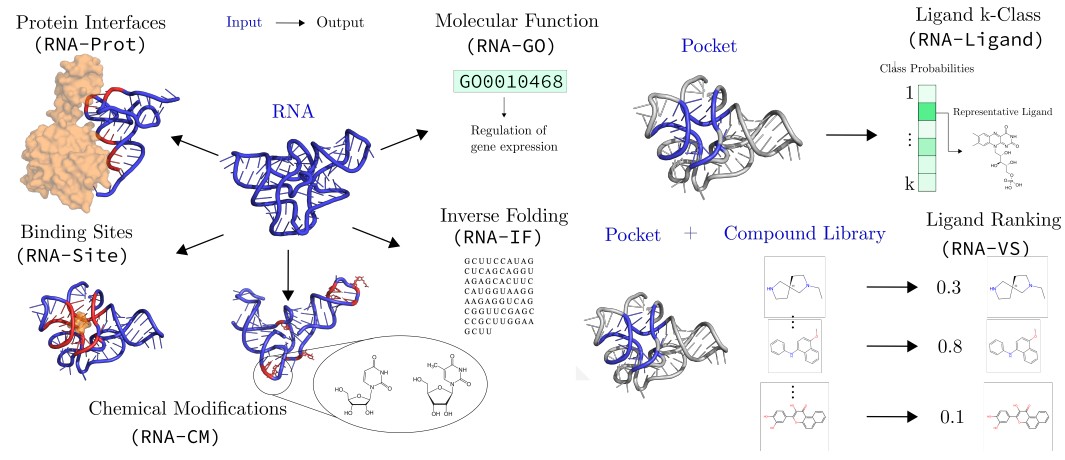

| TASK NAME | SHORT DESCRIPTION |
|---|---|
| 1. RNA-GO | Classify RNAs into one or more frequent function. |
| 2. RNA-IF | Find a sequence that folds into this structure. |
| 3. RNA-CM | Predict which RNA residues are chemically modified from its geometry. |
| 4. RNA-PROT | Predict which RNA residues are interacting with a protein. |
| 5. RNA-SITE | Predict which RNA residues are interacting with a small-molecule protein. |
| 6. RNA-LIGAND | Classify an RNA binding site, based on the ligand it binds to. |
| 7. RNA-VS | Predict small molecule-RNA binding site affinity, and use it for Virtual Screening (VS). |

Figure 1: A graphical overview of the seven tasks included in our library, illustrated by a precursor tRNA (4GCW) and a SAM-I riboswitch (5FJC). The integrated table contains short task summaries.

## 4.1 RNA-GO: FUNCTION TAGGING

**(Definition)**: This is an RNA-level, multilabel classification task where an RNA is mapped to five possible labels representing molecular functions.

**(Context)**: Function-prediction models have the capacity to uncover new structure-function connections. The Gene Ontology (GO) (Ashburner et al., 2000) was developed to associate a function with a gene, and thereby to the RNA or protein it encodes. It resulted in functional categories called *GO terms*, whose prediction from a protein structure was proposed as a task in (Gligorijević et al., 2021), which is now regularly used. A manual annotation of RNAs with GO terms is available in Rfam (Griffiths-Jones et al., 2003; Ontiveros-Palacios et al., 2025), a database of non-coding RNA families.

**(Processing)**: GO terms corresponding to an RNA structure are extracted, then filtered based on their frequency, removing the two most frequent labels (ribosomal systems and tRNAs), and infrequent ones (fewer than 50 occurrences). We group together GO-terms that are fully correlated, resulting in five classes. Finally, we discard RNA fragments with fewer than 15 residues, resulting in 499 systems. We split the data along sequence similarity, following Gligorijević et al. (2021).

## 4.2 RNA-IF: MOLECULAR DESIGN

**(Definition)**: This is a residue-level, classification task. Given an RNA structure with masked sequence information (keeping only the coordinates of the backbone), we aim to recover the sequence.

**(Context)**: One avenue for designing new macromolecules is to fix a coarse-grained tertiary structure first, for instance scaffolding a structural motif, and then predict a sequence that will fold into this given structure. This second step is called inverse folding (IF), since it maps a known structure to an unknown sequence. Inverse folding is a well-established task in protein research with many recent breakthroughs (Watson et al., 2023), with adaptations for RNA with classical models (Leman et al., 2020a). Learning-based approaches were pioneered by *RDesign* (Tan et al., 2023), *gRNAde* (Joshi et al., 2024), while other papers specifically address backbone generation (Anand et al., 2024) and protein-binding RNA design (Nori and Jin, 2024).

**(Processing)**: We gather all connected components in our database between 15 and 300 nucleotides, then remove identical sequences using CD-HIT redundancy removal with a threshold of 0.9 and

split on structural similarity at a threshold of 0.5. We also provide datasets and splits from *gRNAde* (Joshi et al., 2024). Our dataset differs by allowing multichain systems, a stricter size cutoff, and a duplicates filter, leading to a reduced dataset size.

### 4.3 RNA-CM: DYNAMIC FUNCTION MODULATION

(**Definition**): This is a residue-level, binary classification task where given an RNA structure, we aim to predict which, if any, of its residues are chemically modified.

(**Context**): In addition to the four canonical nucleotides, more than 170 modified nucleotides have been found in RNA polymers (Boccaletto et al., 2022), affecting the functions of a diverse range of ncRNAs (Roundtree et al., 2017). We propose to detect such chemical modifications from subtle perturbations of the canonical geometry of the RNA backbones.

(**Processing**): We start with the entire dataset, partition it into connected components and apply our size filter. Then, we filter for systems that include modified residues, relying on PDB annotations. We apply our default redundancy removal and splitting strategies, which results in 197 data points.

### 4.4 RNA-PROT: BIOLOGICAL COMPLEX MODELING

(**Definition**): This is a residue-level, binary classification task where given an RNA structure, we aim to predict whether each residue is within 8Å of a protein residue.

(**Context**): RNAs and proteins often bind to form a functional complex. Such complexes are involved in crucial cellular processes, such as post-transcriptional control of RNAs (Glisovic et al., 2008). RNA structure is often, but not always, heavily disrupted upon interaction with a protein.

(**Processing**): To build this task, we start from the `bgsu` non-redundant dataset, partition it into connected components and apply our size and resolution filters. We remove systems originating from ribosomes that are fully encapsulated within a complex. Then, we retain only RNA interacting with a protein and apply our default redundancy removal and splitting strategies, yielding 1251 data points.

**Tasks for RNA Drug Design**

RNA is increasingly recognized as a promising class of targets for the development of novel small molecule therapeutics (Falese et al., 2021; Haga and Phinney, 2023; Disney, 2019; Abulwerdi et al., 2019). The ability to target RNA would drastically increase the druggable space, and propose an alternative for overused protein targets in scenarios where they are insufficient. RNAs represent a therapeutic avenue in pathologies where protein targets are absent, e.g. triple-negative breast cancer (Xu et al., 2020). The following tasks address questions in RNA small-molecule drug-design.

### 4.5 RNA-SITE: DRUG TARGET DETECTION

(**Definition**): This is a residue-level, binary classification task where given an RNA structure, we aim to predict whether each residue is closer than 6Å to a ligand.

(**Context**): The classic structure-based drug-discovery pipeline starts with the identification of relevant binding sites, which are parts of the structure likely to interact with ligands, or of particular interest for a phenotypic effect. The binding site structure can then be used to condition the search for small molecule binders, for instance using molecular docking (Ruiz-Carmona et al., 2014). The framing of this problem as a machine-learning task for RNA was introduced in *Rbind* (Wang et al., 2018).

(**Processing**): We provide datasets and splits from *RNASite* (Su et al., 2021), which are also utilized by other tools such as *RLBind* (Wang et al., 2023). This dataset contains 76 systems obtained after applying stringent clustering on an earlier version of the PDB. In addition to the published dataset, we propose a larger, up-to-date dataset. We start by following similar steps as for RNA-Prot (without removal of ribosomal systems), then apply a drug-like filter on the small-molecules. Finally, we remove systems with more than 10 protein atom neighbors, ensuring that RNA modulated binding. The default redundancy removal and splitting results in 224 systems.

### 4.6 RNA-LIGAND: POCKET CATEGORIZATION

(**Definition**): This is a binding site-level, multi-class classification task where the structure of an RNA binding site is classified according to the ligand it binds.

**(Context)**: Equipped with a binding site, one wants to use its structure to characterize its potential binders. On proteins, the *Masif-Ligand* tasks (Gainza et al., 2020) gathers all binding sites bound to the seven most frequent cofactors, and aims to classify them based on their ligands. Inspired by this work, we propose the *RNA-Ligand* task. To retain sufficiently many examples per task, we only retained the three most frequent classes. This task is less ambitious than training a molecular docking surrogate and can help understanding the potential modulators of a given binding site.

**(Processing)**: Starting with steps similar to RNA-Site, we obtain a set of structures that display RNAs in interaction with one or more drug-like small molecules. We then extract the context of the binding pocket by seeding two rounds of breadth-first search with all residues closer than 8Å to an atom of the binder. This results in binding pockets, which we group by sequence clustering. To find the most frequent ligands binding in non-redundant pockets, we discard binding sites that bind to several ligands and retain only binding events to the three most frequent ligands in the remaining set. We split this final set based on structural similarity.

## 4.7 RNA-VS: DRUG SCREENING

**(Definition)**: This is a binding site-level regression task. Given the structure of an RNA binding site and a small molecule represented as a molecular graph, the goal is to predict their binding affinity.

**(Context)**: Beyond fixed-category classification, virtual screening aims to score compounds based on their affinity to a binding pocket. This task is ubiquitous in drug design as it helps select promising compounds to be further assayed in the wet-lab. Here, we implement the virtual screening task introduced in (Carvajal-Patiño et al., 2025). Their model is trained to approximate normalized molecular docking scores, and can then rank compounds by their binding likelihood to target sites.

**(Processing)**: The dataset is reproduced from the RNAmigos2 (Carvajal-Patiño et al., 2025) paper. The authors curated a list of binding sites analogously to the RNA-Site task and clustered the sites using RMScore (Zheng et al., 2019). All binders found for each cluster are retained as positive examples, and a set of drug-like chemical decoys are added as negative partners. Molecular docking scores are computed with rDock (Ruiz-Carmona et al., 2014) on all binding site-small molecule pairs.

## 5 IMPLEMENTATION

Next, we briefly showcase the use of our framework for a simple programmatic access to proposed tasks. In Figure 2, we show how practitioners can access our datasets from Python code, automatically downloading them from Zenodo, choose a representation (in this example a *Pytorch Geometric* graph) and directly use the data in a simple and reproducible fashion. Additionally, by passing a single flag (`recompute=True`) to the task fetcher, users can choose to execute all processing logic from scratch, ensuring end-to-end reproducibility.

```
1  from rnaglib.tasks import get_task
2  from rnaglib.representations import GraphRepresentation
3
4  task = get_task(task_id='rna_site',root='example')
5  task.add_representation(GraphRepresentation(framework="pyg"))
6  task.get_split_loaders()
7
8  for batch in task.train_dataloader:
9      rna_graph = batch["graph"]
10     target = rna_graph.y
```

Figure 2: Obtaining a machine learning-ready split dataset requires only a few lines of code.

Due to the rapid advances in the field, we can expect that additional interesting challenges will arise in the near future, complementary to the seven tasks introduced here. Thanks to the modularity of our tool, additional tasks on RNA can be easily integrated in our framework for future releases. This is illustrated in the documentation at `rnaglib.org`[2].

---

[2]Identifiable information relates to the base package, not our task library.

# 6 EXPERIMENTS AND RESULTS

We explicitly limit the scope of this work to the creation of a useful deep-learning library for RNA 3D structure based modeling. We refrain from suggesting novel model architectures and instead aim to illustrate the practical applicability of our library. Still, we provide a baseline performance for each of our proposed tasks as a reference for future works and demonstrate our library's use in understanding and building RNA structure-function models.

All reported experiments were conducted over three seeds, to report error bars in our plots. Implementation details are provided in Section F. The code to reproduce our model training and our plots is available from anonymous.4open.science/r/rnaglib-experiments-1F7B.

## 6.1 DATA ANALYSIS AND TIMING

We characterize our tasks by analyzing the distribution of RNA families within them, primarily using Rfam (Ontiveros-Palacios et al., 2025), the RNA equivalent of Pfam (Mistry et al., 2021). However, due to a significant number of unannotated samples in Rfam, we also use the Nucleic Acid Knowledgebase (NAKB) (Lawson et al., 2023), the intended successor to the Nucleic Acid Database (NDB) (Coimbatore Narayanan et al., 2014), which provides broader coverage and additional nucleic acid–specific annotations. Across all tasks, we find a high number of tRNAs and rRNAs, which mirrors the distribution of RNA structures in the Protein Data Bank (PDB). A complete breakdown of these family and size distributions is available in Appendix C. Despite incomplete annotations, our datasets showcase the variety of functional roles played by RNAs, and thereby illustrate the suitability of our tasks for training models generalizable across families and molecular sizes.

Training a model on a task can be computationally demanding and thus a potential bottleneck. Given the modest amount of RNA structural data, we show that training on our tasks is feasible with limited compute, making the suite accessible to a wide variety of users. Concrete training times and memory requirements for each task are reported in Appendix D.

## 6.2 REDUNDANCY AND SPLITTING

Biological data points are inherently correlated due to evolutionary history, topological similarities, and minor experimental variations that introduce redundancy. Consequently, this data is not independent and identically distributed (i.i.d.). This non-i.i.d. nature means that commonly used random data splits can be insufficient (Volkov et al., 2022), which has motivated the development of more advanced splitting strategies (Bernett et al., 2024).

The choice of splitting strategy depends on the desired type of generalization, from simple interpolation to the more challenging extrapolation across dissimilar proteins or RNAs. The latter is driven by the ambition to use machine-learning to approximate the universal physical laws that are believed to govern structure-function relationships (Nori et al., 2025). To this end, similarity-aware evaluation allows researchers to precisely tune the desired degree of generalization, bridging the gap between random and strict similarity-based splits (Zhang et al., 2025).

To showcase our library's utility in choosing suitable splitting approaches, we compare its three main splitting approaches in Figure 3. As expected, random splitting often leads to inflated performance metrics. However, this effect was less pronounced on our datasets, which we attribute to our initial redundancy removal step mitigating data leakage. Indeed, forgoing the redundancy removal for three example tasks markedly increases the performance gap between similarity-based and random splitting approaches.

For each splitting strategy, our library permits continuous tuning of the similarity thresholds, depending on use case and data availability. The trade-off between leakage and learning signal can thereby be tweaked by practitioners. In Supplementary Figure 7, we plot the performance of our baseline model on the redundant version of RNA-Site, varying the similarity threshold used. Again, a more stringent splitting leads to a decreased performance.

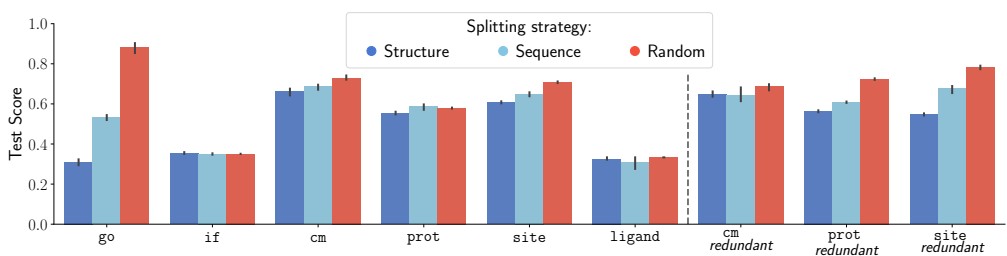

Figure 3: Redundancy removal has a dampening effect on data leakage across splitting strategies.

## 6.3 MODEL BENCHMARKING

We provide two case studies on selected tasks elucidating how our library can assist in choosing modeling modalities. We investigate the impact of increasing the depth of our model, as well as the effect of using different RNA representations. We focus on RNA-Site, RNA-CM, and RNA-Prot.

We evaluated three classes of RNA representations reflecting its multi-level structure. 1D sequence models, such as LSTM or Transformers, use only the raw nucleotide sequence.

2D coarse-grained graphs represent nucleotides as nodes and encode structural information as edges. These range from simple base-pair and backbone connections considered as one (2D) or several edge types (2D+) to the 2.5D representation, which adds non-canonical interactions using Leontis-Westhof nomenclature (Leontis and Westhof, 2001). Those graphs are processed by a Relational Graph Convolutional Network (RGCN).

Finally, 3D geometric graphs explicitly model tertiary structure using residue coordinates, along with k-nearest neighbors edges (GVP) or with the 2.5D representation edges (GVP-2.5D). These graphs are processed by an equivariant graph neural network, the Geometric Vector Perceptron (GVP) Jing et al. (2020). Further details about the representations and models are provided in Appendix E.

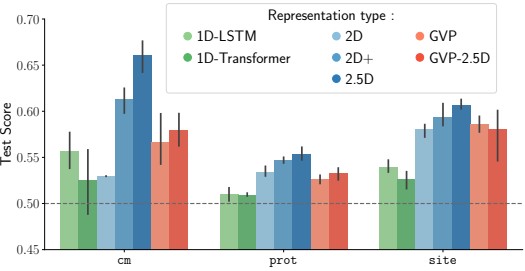

Figure 4a: Performance of different RNA representations: sequence (green), coarse-grained (blue) or full 3D (red).

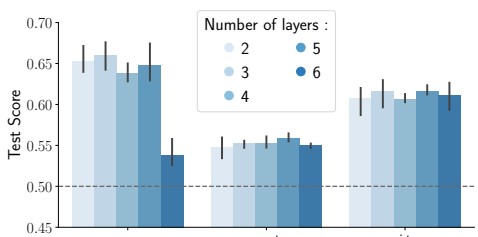

Figure 4b: Performance of the coarse grained approach for different RGCN depths.

Our results clearly indicate that coarse-grained models outperform 3D methods (Figure 4a). We hypothesize that atomic-level detail is unnecessary for modeling inherently unstable RNA structures, particularly with limited data. This is in line with existing findings on predicted structure Xu et al. (2024). However, our results also show that 3D methods outperform 1D methods, which contrast with the results on predicted structures, where noisy 3D information offered no advantage over sequence data. Our use of experimental structures demonstrates the definitive benefit of incorporating 3D structural information over 1D models.

Moreover, our results demonstrate that base-pairing information is a crucial prior for both 2D and 3D models, consistent with previous results obtained on a virtual screening task on graph representations Oliver et al. (2020). The 2.5D representation was the top performer, likely due to its combination of a powerful biological prior (non-canonical pairs) with an efficient, sparse graph structure.

Finally, we investigated the effect of structural context size by varying the depth of our 2.5D network from two to six layers (Figure 4b). While four layers consistently represented a strong baseline, the

optimal depth is task-dependent. For instance, predicting protein binding sites (RNA-Prot) appears to require less structural context than other tasks.

## 6.4 BASELINE PERFORMANCE

Building on those results, we use our best-performing 2.5D approach for each of our proposed tasks to serve as a baseline. We report our results in Table 1 and additional details on the experimental setup, including hyperparameters, and further metrics are in the supplementary Section G. We also include the state-of-the-art performance for existing tasks. Furthermore, we compare our RGCN approach to existing methods on the TR60/TE18 dataset split (Su et al., 2021) for the *RNA-Site* task and the dataset and splits from Joshi et al. (2024) for the *RNA-VS* task. Performance metrics are in Supplementary Tables 7 and 8.

Table 1: Baseline performance reported using our recommended representative metric for each task. We compare to always predicting the majority class. Scores with an asterisk refer to existing literature tasks and are reported directly from their respective publications.

| Task | Metric | Majority | Score | Dataset split sizes |
|------|--------|----------|-------|---------------------|
| RNA-GO | Jaccard | 0.2 | $0.533 \pm 0.013$ | 349-75-75 |
| RNA-IF | Sequence Recovery | 0.28 | $0.354 \pm 0.005$ | 1700-448-581 |
| *gRNAde* | Sequence Recovery | 0.28 | *0.568** | *11183-528-235* |
| RNA-CM | Balanced Accuracy | 0.5 | $0.661 \pm 0.017$ | 138-29-30 |
| RNA-Prot | Balanced Accuracy | 0.5 | $0.553 \pm 0.007$ | 881-189-189 |
| RNA-Site | Balanced Accuracy | 0.5 | $0.607 \pm 0.005$ | 157-34-33 |
| *RNASite* | AuROC | 0.5 | *0.703** | *53-6-17* |
| RNA-Ligand | AuROC | 0.33 | $0.650 \pm 0.006$ | 203-43-44 |
| RNA-VS | AuROC | 0.5 | $0.759 \pm 0.005$ | 304-34-65 |

Despite using a significantly simpler model, our approach achieves competitive results on those tasks, aligning with recent methods while falling short of state-of-the-art performance. We observe that on all tasks, our simple machine learning models are able to capture a meaningful signal that widely outperforms a naive approach of predicting the majority class. This confirms that our benchmark tasks are both learnable and challenging, providing a robust foundation for future model development.

## 7 DISCUSSION

We have introduced a versatile and modular library designed to advance deep learning for RNA structural analysis. By standardizing datasets, splits, and evaluation as benchmarking tasks, the library enables reproducible comparisons and lowers the barrier for developing new computational models. Its standardized workflows foster confidence in results, and its modularity makes it straightforward for researchers to extend the suite with additional tasks.

Our experiments show that RNA tasks are both learnable and challenging, and that representation choice matters: coarse-grained 2.5D graphs provide strong baselines, while 3D encoders capture additional signal beyond sequence-only models. These findings offer practical guidance for model development and underscore the importance of well-founded splitting strategies.

Looking forward, the benchmark can be extended to new directions such as structural model assessment (Townshend et al., 2021c), transferable RNA embeddings, and the integration of dynamic conformational ensembles. The built-in 2.5D representations have already enabled competitive RNA models (Wang et al., 2024) and hold further promise, while the modular design ensures that new and improved representations can be seamlessly integrated as they emerge.

By consolidating diverse RNA structure–function challenges into a single, extensible library, we create an accessible common foundation for future research in the field. We hope that this benchmark will facilitate the development of new deep learning tools in the highly promising area of RNA structural biology.

## REPRODUCIBILITY STATEMENT

All results can be reproduced with the code provided in the experiments repository. All plotting scripts are also available, along with precomputed results files. We also provide two simple scripts to retrain all models.

The code to use the benchmark and the proposed tasks is extensively documented. A multitude of tutorials and how-tos exist with regards to all parts of the package. The tool can be easily installed as a python package and is maintained on Github. Additional and complete information on preprocessing is available in Appendix B. Results can be reproduced with limited compute as specified in Appendix D.

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

# APPENDIX

In this Appendix, we first clearly differentiate the contribution of the current work with the previously published `rnaglib` in Section A. We then provide analysis of the initial data, a justification of the generic data preprocessing, and details of extra-steps taken for specific tasks (Section B). In Section C, we provide a data analysis of the resulting datasets obtained for each of our tasks. In Section D, we analyze computational requirements of training models on our benchmarks. In Section F, we report additional details about the representation and models used in the paper, as well as about the training parameters used for the models presented in the main body of the paper. Finally, we provide additional results of the models tested on our tasks (Section G).

## A POSITIONING WITH REGARDS TO RNAGLIB

The proposed tool is integrated within a pre-existing `rnaglib` python package (Mallet et al., 2022). In this Section we expand on the differences between the two.

The pre-existing package consists of a database of pre-computed 2.5D graphs, lowering the barrier to entry for people interested in using this representation in their applications. In addition, it proposed a set of methods to compare 2.5D graphs using a customized graph-edit distance, plotting scripts, and a kernel to compare 2.5D rooted subgraphs. Finally, it proposed a machine-learning model compatible with this representation, with metric-learning pretraining on the kernels, as proposed in RNAmigos Oliver et al. (2020).

Our current submission significantly expands on these fundamentals. Notably, we introduce all the infrastructure necessary to build a task. We introduce feature computers to easily integrate additional features and annotations, such as RNA language model embeddings. We create a representation abstraction and collating routing to go beyond the exclusive use of 2.5D representation. We also add a dataset object that can be saved or downloaded from Zenodo, as well as routines to remove redundancy and split a dataset. All of these abstract objects are designed to be flexible and modular so that implementing a representation or splitting strategy smoothly fits in our pipeline.

We then implement concrete tools using this infrastructure. Examples include GO term annotators, USAlign splitters or support for sequence, 3D point clouds, voxel grids or 3D graphs representations, and are detailed in Section 3. Finally, we propose our benchmark tasks taking advantage of those tools (see Section 4), as well as results leveraging this benchmark (see Section 6).

## B DATASETS PROCESSING

### B.1 GENERAL PREPROCESSING

In this Section, we provide additional insights about the data and the way they are being preprocessed.

As mentioned in Section 3, RNA molecules in the PDB files exhibit a bimodal distribution over the number of residues. Figure 5a displays the distribution of the RNA sizes (defined as the number of residues) of the RNAs from the PDB.

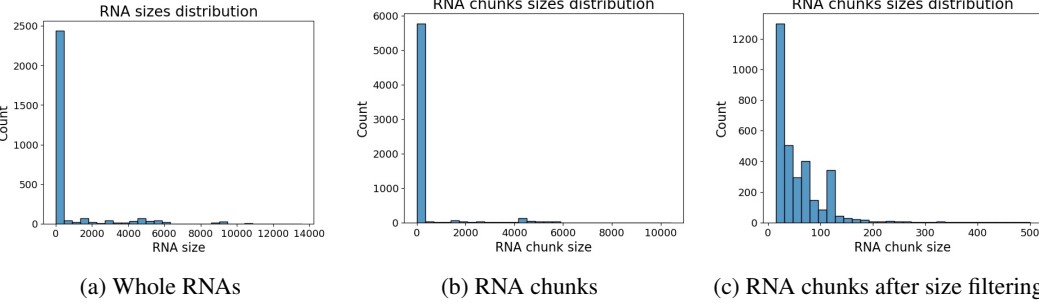

(a) Whole RNAs       (b) RNA chunks       (c) RNA chunks after size filtering

Figure 5: Distribution of RNAs or RNA chunks sizes

A single system, such as rRNA, can include several independent, non-interacting RNA chains in a single PDB file. In order to work at a more biologically relevant scale, we partition the raw RNAs from PDB files into connected components. Figure 5c shows the distribution of the number of residues by connected components.

We remove the RNA components of insufficient size for structure-based machine learning, as well as very large components which negatively impact the computational performance of data loading as well as model training. To this end, we filter RNA chunks and only keep those with between 15 and 500 nucleotides. Figure 5c displays the distribution of RNA chunks sizes after filtering.

Our redundancy-removal process is as follows. First, we perform sequence-based clustering based on a similarity threshold above which RNA fragments are clustered together (using the sequence similarity metric computed using *CD-HIT* (Fu et al., 2012) clustering program). Then, within each cluster, we select the sample with the highest resolution. We then perform structure-based clustering and structure-based redundancy removal (relying on the TM-Score, a structural similarity metric, computed using US-align platform (Zhang et al., 2022a)) following the same procedure. When instantiating the splitters, we perform structure-based clustering with a different threshold to define the clusters which will be required to be grouped either in train, val or test set (which we call "splitting clustering"). Since we only select one representative sample per cluster, the number of clusters determines the number of samples of the final dataset. Therefore, a tradeoff is to be considered between having a large amount of data and discarding redundant RNAs.

For this reason, we study the impact of both redundancy removal and splitting clustering threshold on the number of clusters generated in the case of the RNA-CM task. Results are reported in Figure 6. Figure 6a displays, for 4 different sequence-based redundancy removal thresholds (each with a different color), the scatter plot of the number of clusters obtained based on the structure-based splitting clustering threshold. Here, the redundancy removal thresholds 0.90, 0.80 and 0.70 give the same plot since CD-HIT similarity values are strongly concentrated around 0.5 and 1. The chosen value is 0.90. Figure 6b displays, for 4 different structure-based redundancy removal thresholds (each with a different color), the scatter plot of the number of clusters obtained based on the structure-based splitting clustering threshold. In all cases, the structure-based redundancy removal is performed after a sequence-based redundancy removal using a 0.90 threshold. In all experiments, the splitting clustering is performed based on US-align structural similarity.

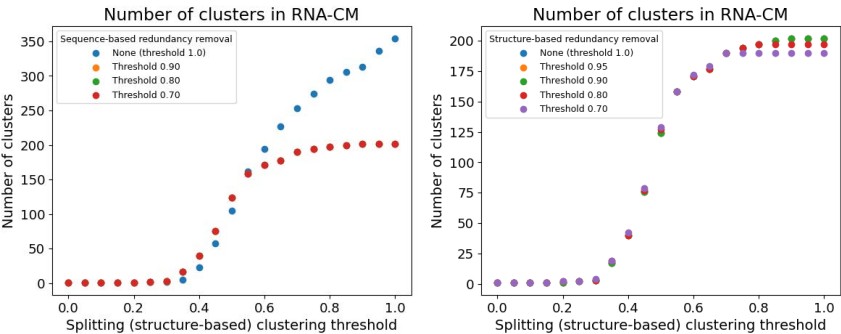

(a) Using sequence-based redundancy removal

(b) Using structure-based redundancy removal

Figure 6: Number of clusters after similarity clustering based on the splitting clustering threshold

Our library permits setting different structure and sequence similarity thresholds for splitting, depending on use case and data availability. In Figure 7, we plot the performance of our baseline model on the redundant version of RNA-Site, varying the similarity threshold used. Our results show that more stringent splitting leads to a decreased, more realistic performance. The trade-off between leakage and learning signal can thereby be tweaked by practitioners. We point out that structure and sequence thresholds are not directly comparable, since their calculation differs. Setting a threshold of 1 results in isolated points for our structure similarity metric, resulting in comparable results to random splitting. This is in contrast with our sequence-based similarity metric, whose value is exactly

one for highly similar structures. Using either a non-redundant dataset or rigorous splitting, alleviates the risk of data leakage, and using both maximizes the reliability of the results.

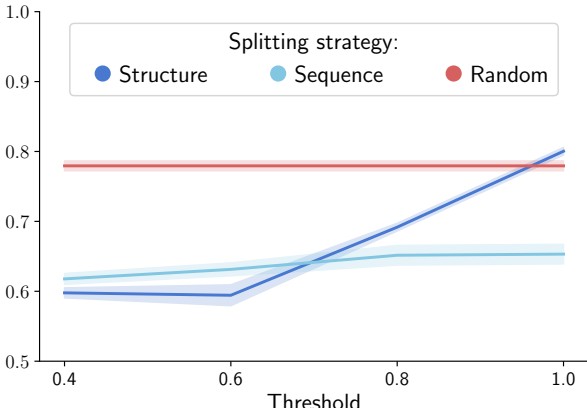

Figure 7: In a redundant setting, stricter splitting thresholds reduce data leakage, shown for RNA-Site.

## B.2 Task-specific Preprocessing

We provide additional details about the tasks involving a specific data preparation process.

**RNA-Ligand** We selected the three most frequent ligands that were neither modified RNA or DNA residues nor modified protein residues, following the ligand definition proposed by *Hariboss* (Panei et al., 2022). We only retained the binding pockets binding to one of these three ligands in order to ensure the amount of binding pockets binding to each ligand would be significant enough to enable learning in a multi-class classification framework. The three ligands retained are paromomycin (called PAR in the PDB nomenclature), gentamicin C1A (LLL) and aminoglycoside TC007 (8UZ). Their structures are displayed in Figure 8.

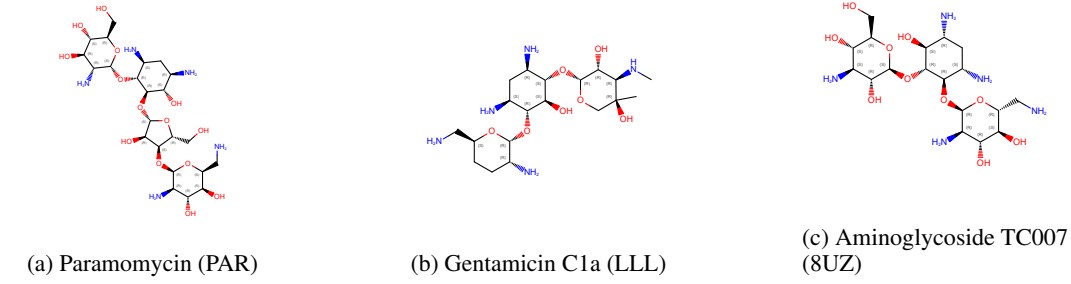

(a) Paramomycin (PAR)     (b) Gentamicin C1a (LLL)     (c) Aminoglycoside TC007 (8UZ)

Figure 8: Structures of the ligands selected for the RNA-Ligand task

**RNA-GO** When building RNA-GO, we explore all the GO-terms of RNAs from the PDB, remove the GO-terms which occur more than 1000 times (these are in fact the GO-terms corresponding to "structural constituent of ribosome", "ribosome" and "tRNA"). We also remove the GO-terms which are underrepresented (those which occur less than 50 times in our experiments). We then remove the GO-terms which are very correlated to other ones by performing GO-terms clustering based on correlation matrix (with correlation threshold 0.9) and keeping only one representative GO-term per cluster. After this preprocessing, 5 GO-terms are remaining: 0000353 (formation of quadruple SL/U4/U5/U6 snRNP), 0005682 (U5 snRNP), 0005686 (U2 snRNP), 0005688 (U6 snRNP) and 0010468 (regulation of gene expression).

## C  DATA ANALYSIS

Figures 9–10 show the family distributions of our proposed datasets, annotated with both Rfam and NAKB labels. We have chosen Rfam clans and NAKB annotations as the right level of granularity, since they provide a good overview of the present functions, without being overly detailed or too general. Across tasks we observe substantial functional diversity, with the exception of RNA-Ligand, which was pre-processed to include only RNAs known to bind one of three ligands. Overall, NAKB provides slightly broader coverage of annotated RNA structures compared to Rfam. Importantly, the Rfam-annotated RNAs are not a strict subset of those annotated by NAKB, so the true combined coverage is somewhat higher. To our knowledge, however, there is no standardized mapping between Rfam clans and NAKB annotations.

A large fraction of the structures correspond to ribosomal complexes, particularly ribosomal RNAs and tRNAs. This bias reflects the composition of the PDB, where many RNA crystal structures originate from RNAs crystallized in complexes with proteins. Given that the ribosome is a central molecular machine of the cell and has been extensively studied by structural biologists, its overrepresentation in our data is expected.

Table 2 reports dataset statistics, highlighting the wide range of RNA sizes present in our collection. Taken together, the diversity of RNA functions and sizes represented in our datasets ensures that models trained on them are likely to generalize across RNA types.

Table 2: We provide statistics on the size distribution of RNA samples in our datasets. Note that depending on the task, a size cutoff at either 300 or 500 nucleotides is applied.

| Task Name | Total Nodes | Dataset Size | Min Nodes | Max Nodes | Mean Nodes | Median Nodes | RNAs $\geq 200$ | RNAs $\geq 200$ (%) | Num Classes | Num Features |
|---|---|---|---|---|---|---|---|---|---|---|
| rna_cm | 11,208 | 197 | 16 | 482 | 56.89 | 40.00 | 4 | 2.0 | 2 | 4 |
| rna_go | 46,557 | 499 | 10 | 299 | 93.30 | 93.00 | 3 | 0.6 | 5 | 4 |
| rna_if | 164,769 | 2,729 | 16 | 298 | 60.38 | 44.00 | 72 | 2.6 | 5 | 1 |
| rna_ligand | 14,632 | 290 | 27 | 78 | 50.46 | 49.00 | 0 | 0.0 | 3 | 4 |
| rna_prot | 74,257 | 1,274 | 16 | 488 | 58.29 | 40.50 | 36 | 2.8 | 2 | 4 |
| rna_site | 14,430 | 224 | 16 | 478 | 64.42 | 47.00 | 9 | 4.0 | 2 | 4 |

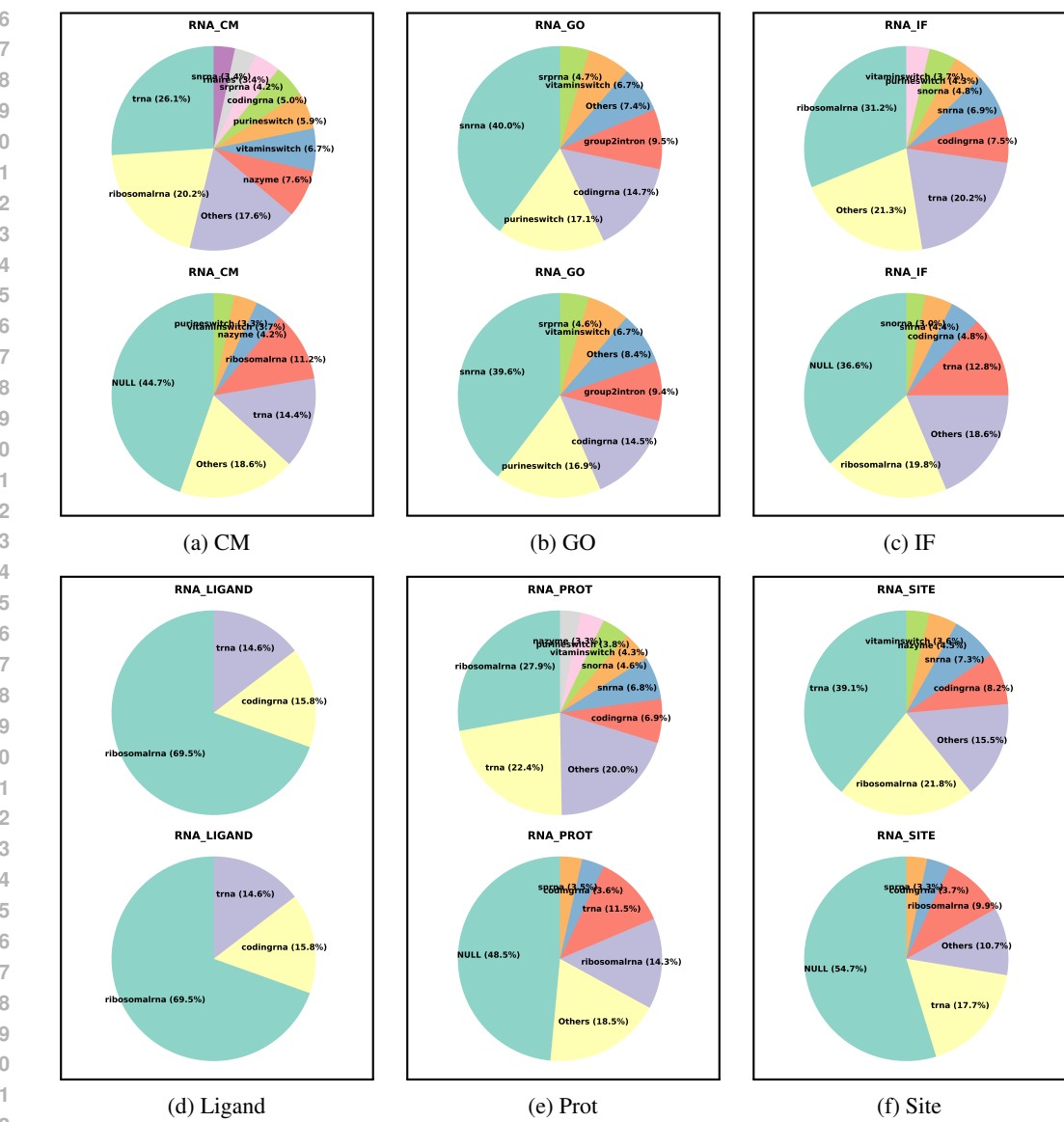

Figure 9: Distribution of NAKB functional annotations across task datasets. For each dataset, the top plot excludes unannotated RNAs and the bottom includes them.

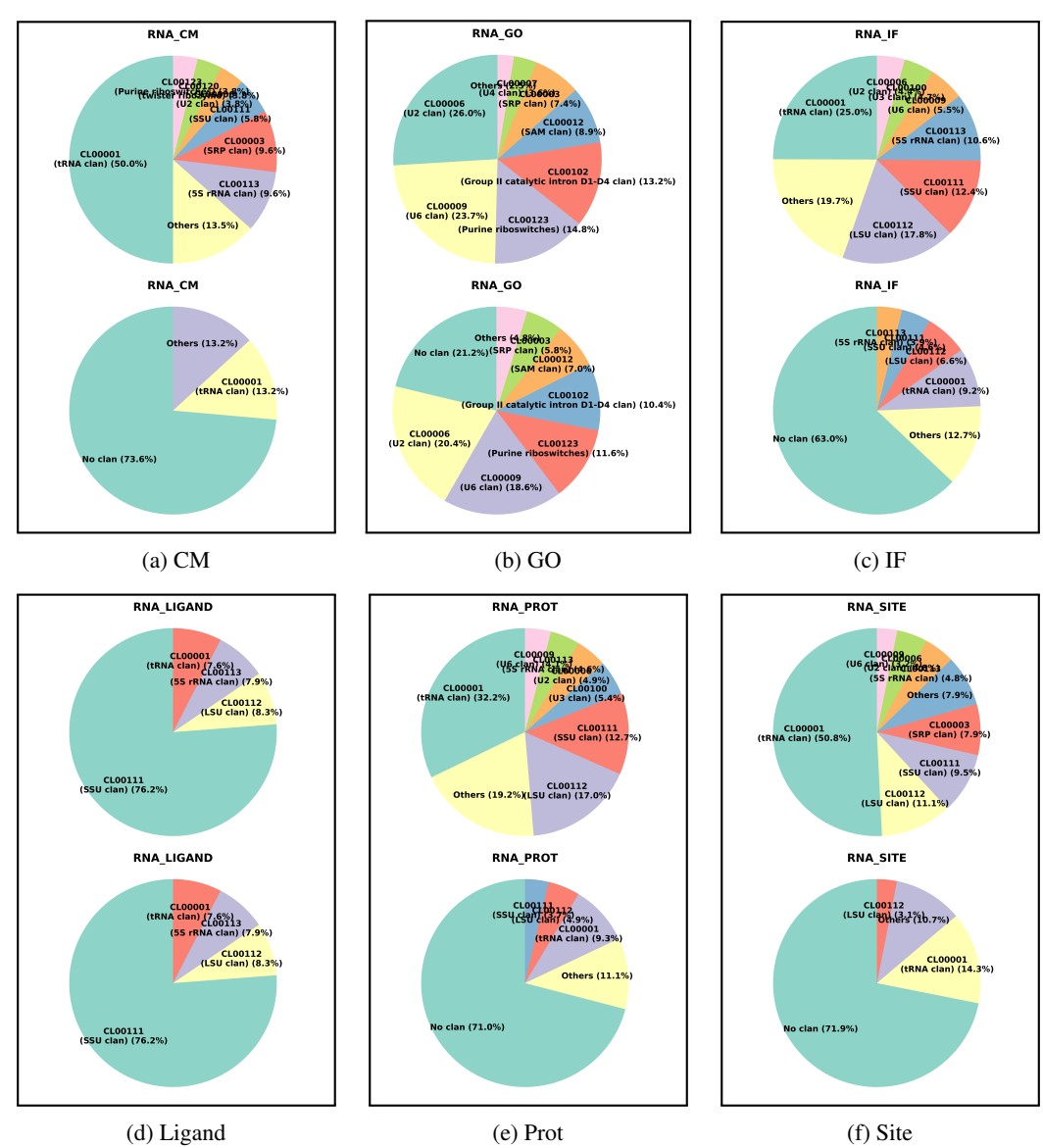

Figure 10: Distribution of Rfam clans across task datasets. For each dataset, the top plot excludes unannotated RNAs and the bottom includes them.

## D  TIMING

We ran our proposed models with a batch size of 8 on 4 Intel i9 CPU cores or on a personal GPU (RTX 3500), and report runtimes and memory requirements in Table 3. Given the modest dataset sizes, runtimes and memory use were not extensively optimized but are unlikely to pose practical limitations. Depending on the task, training a model takes between a few minutes and one hour. Training on a GPU results in a twofold speedup.

Table 3: An overview of the time and memory requirements of training our models for each provided task.

| Task Name | Time/RNA (CPU ms) | Time/RNA (GPU ms) | Peak GPU Memory (MB) |
|-----------|-------------------|-------------------|----------------------|
| RNA_CM | 22.4 | 9.69 | 60.66 |
| RNA_Go | 23.77 | 11.78 | 48.84 |
| RNA_IF | 10.11 | 5.49 | 55.40 |
| RNA_Ligand | 21.46 | 8.65 | 42.00 |
| RNA_Prot | 21.64 | 9.70 | 52.04 |
| RNA_Site | 23.42 | 9.95 | 50.77 |
| RNA_VS | 33.55 | 16.59 | 41.97 |

# E  DETAILS ABOUT REPRESENTATIONS AND MODELS

This section provides a comprehensive overview of the models and RNA representations used in order to foster reproducibility.

## E.1  MODELS

This subsection provides details about the models used.

**Bidirectional Long Short-Term Memory (BiLSTM)** processes sequential data in both forward and backward directions using two separate Long short-term memory (LSTM) layers. Unlike standard LSTMs that only consider past context, BiLSTM captures both past and future information by concatenating outputs from both directions. This bidirectional approach provides richer contextual representations, making it effective for sequence labeling and classification tasks where the complete sequence is available during processing.

**Transformers** are attention-based neural networks that process sequential data in parallel rather than sequentially. The core mechanism is self-attention, which allows the model to weigh the importance of different positions in the input sequence when computing representations for each token. Unlike RNNs, transformers can process all sequence positions simultaneously, enabling faster training and better capture of long-range dependencies. The architecture consists of encoder-decoder blocks with multi-head attention, position encodings, and feed-forward layers.

**Graph convolutional network (GCN)** (Kipf, 2016) is a graph neural network architecture proposing to extend convolution operations to graph data by performing message passing on the 1-hop neighborhood of each node at each layer. The formula of the GCN is the following:

$$h^{l+1} = \sigma(Ah^l W + b)$$

where $h^l$ denotes the embedding at the end of layer $l$, $\sigma$ denotes an activation function, $A$ the adjacency matrix of the graph, $W$ the weights of layer $l+1$ and $b$ the bias of layer $l+1$

**Relational graph convolutional network (RGCN)** (Schlichtkrull et al., 2018) is a refinement of GCN with different edge types, and distinct learnable weight matrices for each edge type. It has the following formula:

$$h^{l+1} = \sigma(W_0 h^l + \sum_{r=1}^{R} W_r A^r h^l)$$

where $R$ is the number of edge types and $A^r$ the adjacency matrix relative to edge type $r$

**Geometric vector perceptron (GVP)** (Jing et al., 2020), initially introduced to learn on protein structures, is an equivariant graph neural network designed to process both scalar and vector node and edge features. In particular, it enables to process 3D coordinates in the learning process. The GVP equations consist in the update equations of both scalar $s^l$ and vector $V^l$ features at layer $l+1$:

$$s^{l+1} = \sigma(W_m[\|W_h V^l\|_2 \| s^l])$$

$$V^{l+1} = \sigma^+(\|W_\mu W_h V^l\|_2) \odot W_\mu W_h V^l$$

where $\sigma$ and $\sigma^+$ are two activation functions and $W_h$ and $W_\mu$ are two learnable linear transformations, $\odot$ denotes row-wise multiplication and $[\cdot\|\cdot]$ denotes concatenation.

### E.2 REPRESENTATIONS

We here detail the various representation frameworks proposed in Rnaglib as Representation objects and used in our experiments.

**2D representation.** RNA is represented as a graph. Each node corresponds to a nucleotide and each edge encodes either a backbone connection or a canonical base pair. Node features are the one-hot encoding of base identity of nucleotides. Edges are not featurized.

**2D+ representation.** RNA is represented as a graph, very similar to the 2D representation. The only difference lies in the presence of three distinct edge types: backbone 3'-5' connections, backbone 5'-3' connections and base pairing edges.

**2.5D representation.** RNA is represented as a graph whose nodes correspond to a nucleotide and edges encode either a backbone connection or a base pair, be it canonical or non-canonical. This representation is built using structures coming from the Protein Data Bank (PDB). Node features encode the base identity of nucleotides and, optionally, comprise an embedding computed from the RNA language model RNA-FM (Chen et al., 2022). This graph has distinct edge types. Not only do these edge types distinguish between backbone and base pair connections (as in 2D+ representation), they also account for the geometry of base pair interactions according to Leontis and Westhof categorization.

**3D representations.** RNA is represented as a graph which nodes represent its nucleotides. We provide two possibilities for edge definition: either a $k$-nearest neighbor graph (edges are built between a node and its $k$ nearest nodes in the 3D space) or a graph which edges encode either a backbone connection or a canonical base pair (as in 2D and 2D+ representation). In our experiments, we report results for $k$-nearest neighbor graphs with $k = 16$. Node scalar features encode the base identity of nucleotides and, optionally, comprise an embedding computed from the RNA language model RNA-FM. Node vector features are unit vectors from nucleotide i-1 to nucleotide i and from nucleotide i to nucleotide i+1) and, optionally, the base pair geometry according to Leontis and Westhof nomenclature (this option is typically used in our *GVP-2.5D* representation). Edge scalar features include radial basis function encoding of the distance between the two nucleotides forming the edge. As edge vector features, we use the unit vector from nucleotide i to nucleotide j, where i and j are the nucleotides represented by the nodes forming the edge. By default and in our experiment, the 3D coordinates of the nucleotides are assumed to be the 3D coordinates of their phosphate atoms. However, the implementation of the representation is very modular and gives the users the possibility to add custom edge or node features, as well as to use different representative atoms to model nucleotide coordinates.

## F TRAINING DETAILS

We provide all the code needed to reproduce our training in the two attached repositories. The code is accompanied by readme documents describing the step-by-step process to reproduce our results. In addition, extensive code documentation is available for all parts of the developed tool. For a simple overview, the most important hyperparameters of the GCNs and RGCNs used in our experiments are summarized in table 4. All our experiments have been performed using a batch size of 8.

Table 4: Hyperparameters used for different RNA tasks in GCN and RGCN models.

| Model | Representation | Layers | Hidden Dim | Epochs | Learning Rate | Dropout |
|-------|---------------|--------|------------|--------|---------------|---------|
| RNA_Ligand | 2.5D | 3 | 64 | 20 | $1 \times 10^{-3}$ | 0.5 |
| RNA_CM | 2.5D | 3 | 128 | 40 | $1 \times 10^{-3}$ | 0.5 |
|  | 2D | 3 | 128 | 40 | $1 \times 10^{-3}$ | 0.5 |
|  | 2D-GCN | 2 | 256 | 40 | $1 \times 10^{-3}$ | 0.5 |
| RNA_Site | 2.5D | 4 | 256 | 40 | $1 \times 10^{-3}$ | 0.5 |
|  | 2D | 2 | 128 | 40 | $1 \times 10^{-4}$ | 0.5 |
|  | 2D-GCN | 4 | 256 | 100 | $1 \times 10^{-3}$ | 0.5 |
| RNA_Prot | 2.5D | 4 | 128 | 40 | $1 \times 10^{-3}$ | 0.2 |
|  | 2D | 4 | 64 | 40 | $1 \times 10^{-2}$ | 0.2 |
|  | 2D-GCN | 5 | 256 | 40 | $1 \times 10^{-2}$ | 0.2 |
| RNA_IF | 2.5D | 3 | 128 | 100 | $1 \times 10^{-4}$ | 0.5 |
| RNA_VS | 2.5D | 3 | 64/32 | 100 | $1 \times 10^{-3}$ | 0.2 |
| RNA_GO | 2.5D | 3 | 128 | 20 | $1 \times 10^{-3}$ | 0.5 |

We report below the hyperparameters used to train GVP models. In the case of GVP models, we don't tune a unique hidden dimension hyperparameter but four: the number of hidden node scalar features $h_s^N$, the number of hidden node vector features $h_V^N$, the number of hidden edge scalar features $h_s^E$ and the number of hidden edge vector features $h_V^E$. In the table below, we call Node Dim the $(h_s^N, h_V^N)$ couple and Edge Dim the $(h_s^E, h_V^E)$.

Table 5: Hyperparameters used for different RNA tasks in GVP models.

| Model | Representation | Layers | Node Dim | Edge Dim | Epochs | Learning Rate | Dropout |
|-------|---------------|--------|----------|----------|--------|---------------|---------|
| RNA_CM | GVP | 3 | (128, 2) | (32, 1) | 40 | $1 \times 10^{-3}$ | 0.5 |
|  | GVP-2.5D | 3 | (32, 2) | (32, 1) | 40 | $1 \times 10^{-3}$ | 0.5 |
| RNA_Prot | GVP | 4 | (32, 2) | (32, 1) | 40 | $1 \times 10^{-3}$ | 0.5 |
|  | GVP-2.5D | 4 | (32, 2) | (32, 1) | 40 | $1 \times 10^{-3}$ | 0.5 |
| RNA_Site | GVP | 6 | (32, 2) | (32, 1) | 40 | $1 \times 10^{-3}$ | 0.5 |
|  | GVP-2.5D | 6 | (32, 2) | (32, 1) | 40 | $1 \times 10^{-3}$ | 0.5 |

For all binary tasks, we apply sample weighting to the loss in order to account for the class imbalance. The weights attributed to positive samples are given by the following formula:

$$\sqrt{\frac{N_{neg}}{N_{pos}}}$$

where $N_{neg}$ is the number of negative samples in the dataset and $N_{pos}$ is the number of positive samples.

For 2D-GCN representations only, we directly use the negative-to-positive ratio as sample weights instead of its square root because we found this reinforcement of the weighting to ease training in this setting.

# G ADDITIONAL RESULTS

## G.1 ADDITIONAL METRICS FOR THE REPORTED MODELS

Table 6: Test performance metrics for various RNA-related tasks. *For RNA_IF, the reported "Global Balanced Accuracy" corresponds to sequence recovery.

| Task | Test F1-Score | Test AUC | Test Global Balanced Accuracy | Test MCC | Test Jaccard |
|---|---|---|---|---|---|
| RNA_Ligand | 0.2771 | 0.6751 | 0.4678 | - | - |
| RNA_CM | 0.1957 | 0.7393 | 0.6615 | 0.1695 | - |
| RNA_Site | 0.3346 | 0.5929 | 0.6309 | 0.3098 | - |
| RNA_Prot | 0.4545 | 0.6654 | 0.6254 | 0.2469 | - |
| RNA_IF | 0.3326 | 0.6201 | 0.3523* | 0.1319 | - |
| RNA_VS | - | 0.855 | - | - | |
| RNA_GO | 0.4074 | 0.8406 | 0.7067 | - | 0.3167 |

## G.2 RESULTS OF THE REPORTED MODELS ON PRE-EXISTING BENCHMARKS

For some of our tasks, versions have already been published in the literature, which is why we include a literature version in addition to our own improved version into our library. While the dataset may be different from ours either in size, or in stringency of splitting criteria, we chose to include them into our tool so that an easy comparison of newly developed models to existing work is possible. To showcase this, we compare some simple models trained with our library on these literature tasks with existing works. While we do not achieve state-of-the-art, our RGCN models are competitive, despite being much simpler and smaller than the published methods. This is an indication of the suitability of graph representations for RNA structure-function modeling, although further research on this is needed.

Table 7: We compare a standard RGCN using the `rnaglib` task module with various published results using the TR60/TE18 split. *Note:* Binding site definitions may vary slightly between models.

| Methods | MCC | AUC |
|---|---|---|
| Rsite2 (Zeng and Cui, 2016) | 0.010 | 0.474 |
| Rsite (Zeng et al., 2015) | 0.055 | 0.496 |
| RBind (Wang et al., 2018) | 0.141 | 0.540 |
| RNASite_seq (Su et al., 2021) | 0.160 | 0.641 |
| RNASite_str (Su et al., 2021) | 0.185 | 0.695 |
| RNASite (Su et al., 2021) | 0.186 | 0.703 |
| *Our library* RNA-Site | 0.113 | 0.607 |

We compare our simple model on the literature RNA-Site task to other published models evaluated on the same data in Table 7. RNASite (Su et al., 2021) combines sequence and structural information based on topological and graph features of the structure into a random forest and thereby yields the best score. Older tools like Rsite2 (Zeng and Cui, 2016) are based on secondary structure only and deliver inferior performance. This is a clear indication that incorporating 3D structural information is essential for solid performance on this benchmark.

Similarly, we also benchmark our simple RGCN architecture on the literature version of the RNA-IF task. There, Joshi et al. (2024) achieve the highest sequence recovery with their model `gRNAde`. `gRNAde` is a geometric deep-learning pipeline using an SE(3) equivariant encoder-decoder. This approach yields higher results as physics based models like Rosetta or the contrastive learning approach RDesign. Remarkably, a simple RGCN performs similarly to the latter and opens doors for further development of graph-based RNA design models. We report the sequence recovery scores in Table 8.

This leads us to conclude that graph representations of RNA offers new opportunities for powerful structure-function models for RNA. We hope that our library will provide the necessary foundation for that development.

Table 8: Sequence recovery scores for RNA inverse folding models. We use a standard two layer RGCN part of `rnaglib`'s task module on the dataset and split published by Joshi et al. (2024)

| Method | Sequence Recovery |
|---|---|
| gRNAde (Joshi et al., 2024) | 0.568 |
| Rosetta (Leman et al., 2020b) | 0.450 |
| RDesign (Tan et al., 2023) | 0.430 |
| FARNA (Alam et al., 2017) | 0.321 |
| ViennaRNA (Lorenz et al., 2011) | 0.269 |
| *Our library* RNA-IF | 0.410 |

## H  LLM USAGE

LLMs were used to assist in coding, writing, and producing figures. All LLM-produced code and text was thoroughly double-checked.

