# OpenReview forum: "A Comprehensive Benchmark for RNA 3D Structure-Function Modeling"
_ICLR.cc/2026/Conference — Submitted to ICLR 2026_

### Official Review · Reviewer_Q7gt · 2025-10-23

**Soundness:** 3
**Presentation:** 3
**Contribution:** 2
**Rating:** 2
**Confidence:** 4

**Summary:**

The paper proposes a benchmarking framework for RNA 3D structure-function modeling. The framework currently includes seven tasks covering diverse biological challenges, each defined by standardized datasets, splitting strategies, and evaluation metrics. The framework is modular, providing reusable components such as annotators, filters, and splitting tools that simplify the creation of new tasks and ensure reproducibility. Finally, the authors establish initial leaderboards by training baseline neural models to evaluate different input representations and splitting schemes. The overall goal is to enable fair, reproducible comparisons and accelerate model development in RNA structural biology.

**Strengths:**

The paper presents a valuable contribution to RNA structural modeling by introducing the first comprehensive benchmark for RNA 3D structure-function prediction. Deep learning is rapidly expanding into bio-related domains, yet suitable datasets and well-defined evaluation pipelines remain fragmented, especially for RNA. By providing standardized datasets, data-splitting strategies, and evaluation metrics for seven biologically diverse tasks, the proposed framework can significantly lower the entry barrier for machine learning researchers entering this field.

The modular design, with ready-to-use annotators and splitting tools, makes it straightforward to add new tasks and ensures reproducibility. The inclusion of simple baselines and initial leaderboards is also valuable, offering an immediate reference point for future model development. Overall, this benchmark has strong potential to accelerate progress by bridging the gap between the ML and RNA-structure communities and making RNA research more accessible to computational scientists.

Overall, the paper is clearly written, and the framework is implemented with attention to reproducibility (open-source code, retraining scripts, and detailed appendices).

**Weaknesses:**

The main limitation of this work lies in its lack of novelty and limited significance for a machine learning audience. The paper does not introduce new modeling techniques, learning paradigms, or analytical insights into ML behavior. Its contribution is primarily infrastructural (data collection, filtering, and standardization), which, while useful, fits better within the scope of bioinformatics resources than an ML research venue. The framework may indeed help computer scientists enter the RNA field more easily, but this convenience comes with a trade-off: by abstracting away biological complexity, it risks encouraging the development of models that rely on poorly understood datasets rather than fostering a deeper understanding of the underlying biology.

**Methodological clarity and justification:**
- Some of the preprocessing and filtering choices were a bit unclear, and it would be beneficial to give more context on how these decisions were made. Lines 119-123 introduce resolution and size cutoffs, but only the upper limit for size is justified, and the rationale for excluding RNAs that depend on protein interactions is not discussed, and this might not be equally important for all tasks. Similarly, the 8 Å cutoff introduced in line 278 is unexplained; if this value is standard in the field, it should be cited, and if not, the authors should justify it empirically. The same applies to the removal of binding sites that bind multiple ligands (line 280), which is presented as a data preprocessing step without any discussion of why such cases are problematic.
- Regarding Figure 5a, I was not entirely convinced by the claim that the size distribution of RNAs is bimodal. From visual inspection, it seems to reflect one dominant group of smaller RNAs and a long tail of very large ribosomal RNAs rather than two distinct peaks. If this interpretation is correct, a short clarification would help avoid confusion about the shape of the dataset distribution.

**Dataset design and labeling:**
- Some aspects of the dataset construction are not entirely clear to me. For example, in the RNA-ligand and RNA-VS tasks, the PDB contains only a limited number of experimentally determined RNA-ligand complexes, so I was wondering how “negative” examples are defined. Does the absence of a complex in PDB imply non-binding, or is there another source used to establish negatives?
- For the RNA-VS task, if I understood correctly, the ground-truth binding scores are generated by rDock. It would be helpful if the authors could comment on how reliable rDock is for RNA systems, and what the motivation is for training models on data derived from this scoring function rather than using the tool directly.
- Looking at Table 1, the train/validation/test split ratios also differ quite substantially across tasks. Could the authors explain why these percentages are not consistent and whether this affects the comparability of results? Finally, the gRNAde dataset appears to contain far more examples (over 11k RNAs) than others, is there a reason why this larger dataset was not incorporated in the main dataset?
- For the RNA-Prot task, I wanted to confirm whether only the RNA sequence or structure is used as input, while the protein partner is not included. If that’s the case, it would be interesting to understand the reasoning behind this setup — doesn’t the identity or nature of the protein influence whether a nucleotide is likely to be in contact? Clarifying this could help readers interpret what kind of signal the model is expected to learn in this task.

**Representation and modeling clarity:**
- I found some parts of the description of input representations and model setups a bit difficult to follow, and additional clarification would be very helpful. In Section 5 and Appendix E, the term “1D representation space” is mentioned, but it was not entirely clear to me what this refers to, I assume it represents the RNA sequence, perhaps via a one-hot encoding, but a short explanation would make this explicit.
- In Figure 4a, I was unsure whether 1D-LSTM and 1D-Transformer use the same input features and differ only in architecture, or if the representations themselves differ in some way. Similarly, in Figure 2, the GraphRepresentation (PyTorch Geometric) seems to correspond to the 3D representation, but this only became clear later in the paper.
- For Figures 4b and Appendix F, it would also help to include the number of parameters or blocks for each model. Since the datasets are relatively small, this context would make the performance comparisons easier to interpret. Finally, I could not find details on how training was terminated (fixed epochs, early stopping, validation criterion, etc.); adding this information would help readers understand how the models were optimized.

**Experimental transparency and reproducibility:**
- I found Section 6.4 and Table 1 informative, but I was not fully sure how to interpret some of the reported results. It would be useful to know which specific model configuration was used for each task; for instance, the number of layers or which representation type was applied, so readers can better understand whether the performance differences come from architecture, data variation, or representation choice.
- The paper mentions reproducibility scripts, which is great, but I could not find details on whether tools such as CD-HIT and US-align were run with default parameters or custom ones. Clarifying this would make the setup easier to reproduce. Additionally, line 305 refers to a “recomputation” option; I was curious what exactly this entails. Does it mean that the datasets can be automatically updated with new PDB entries, or that the benchmark is rebuilt from a fixed snapshot? This would help readers understand how the benchmark can be maintained or extended over time.

**Conceptual limitations:**
- In the discussion, the paper makes relatively strong claims about the advantages of certain representations (for example, that 3D models outperform sequence-based ones), but I was wondering how these conclusions should be interpreted given the limited dataset sizes. There are far more RNA sequences available than experimentally resolved 3D structures, and large-scale sequence pretraining might ultimately prove more effective once transferred to structural tasks. It would be valuable if the authors could comment on this broader perspective, whether they see their results as evidence that 3D structure is intrinsically more informative, or simply as an observation constrained by the current scale of available data.

**Minor (did not influence the decision):**
- Line 27: Please use the full name “AlphaFold 2”. The earlier AlphaFold (2019) model (https://www.nature.com/articles/s41586-019-1923-7) did not have the same impact or recognition, so it’s important to distinguish between the two.
- Line 33: The phrasing suggests that CASP and CAPRI emerged after deep learning advances, while in reality, they date back to 1994 and 2001, respectively, long before neural networks were used in structural biology. It would be good to correct this and cite representative references such as https://onlinelibrary.wiley.com/doi/10.1002/prot.26617 and https://onlinelibrary.wiley.com/doi/10.1002/prot.70018.
- Section 2.2: It would strengthen the context to mention that CASP now includes RNA 3D structure prediction (https://onlinelibrary.wiley.com/doi/10.1002/prot.26550) and that there is a dedicated RNA benchmark, RNA Puzzles (https://www.nature.com/articles/s41592-024-02543-9).
- Section 2.3: It might also be useful to acknowledge recent deep-learning models for RNA 3D structure prediction such as DRfold (https://www.nature.com/articles/s41467-023-41303-9), RhoFold+ (https://www.nature.com/articles/s41592-024-02487-0), and trRosettaRNA (https://www.nature.com/articles/s41467-023-42528-4).
- Line 242: The heading currently looks like it belongs under Section 4.4, but it seems to introduce Sections 4.5 and 4.6 instead, adjusting this could help readability.
- Appendix B.1 (line 866): If I understood correctly, this should reference Figure 5b rather than 5c.
- Figure 6b: The high-similarity thresholds (e.g., 0.95–1.0) seem to overlap or are not visible, and the text doesn’t clarify their position. Annotating or separating these points would make the figure easier to interpret.
- Table 2 (Appendix C): Adding a note that “Nodes = Nucleotides” would make the table easier to understand. I was also unsure what the “number of features” refers to; a short explanation would help.
- References: Several reference titles are not capitalized correctly, this can be fixed by wrapping words in braces ({}) in the BibTeX file.

**Questions:**

1. Some of the datasets included in the benchmark are quite small (for example, 138 or 157 data points in the training set). It would be interesting to hear the authors’ perspective on what they expect can realistically be learned from such limited data.
2. Since the paper includes datasets like gRNAde and RNASite, did the authors consider training their baseline models on these existing datasets to compare performance or confirm consistency across sources?
3. I did not find information about whether it is possible to obtain a list of PDB IDs and corresponding target values for each dataset. Having access to such mappings would be very helpful for quick inspection, dataset validation, and external comparison.
4. Do the authors plan to include state-of-the-art baseline tools or models for each task? I could not find public leaderboards on the documentation website (only Table 1 in the paper). It would be very helpful to have task-specific leaderboards directly available in the online documentation to make performance comparisons and community contributions easier.

---

> ### Author Response · Authors · 2025-11-17
>
> We are very happy the reviewer agrees that our "benchmark has strong potential to accelerate progress by bridging the gap between the ML and RNA-structure communities and making RNA research more accessible to computational scientists." and would like to thank them very much for the extensive feedback provided.
>
> While we agree with most points raised we would like to make a major clarification. We have intentionally submitted to the ICLR datasets & benchmarks category, which to our understanding, is intended for work just like ours, thus we cannot quite agree to the reviewer's suggestion, that a submission to a ML venue is inappropriate, especially given that our benchmarking suite is specifically tailored to the needs of ML practicioners in biology. Given the submission to this category, we feel that the absence of novel modelling techniques is inherent, and thus not a valid criticism of our work. Indeed, poor ready-made benchmarks for ML practicioners abstracts away some of the underlying biological complexity and can lead to spurious results when done incorrectly. This is exactly what motivated us to do this work: by providing benchmarking datasets that are sound from a biological perspective, the work of ML practicioners in improving on our baselines is much more likely to be of biological relevance.
>
> This being said, we are very grateful for all the subsequent and very detailed comments and suggestions being made by the reviewer and will make sure to inlcude them in a next iteration of this work. Here, we propose a brief point-by-point answer.
>
> - The goal of these benchmarks is to model the structure of RNA. Hence, we excluded RNA fragments as they cannot result in complex structures and RNA parts scaffolded by proteins.
> - The 8Å cutoff for binding site is widely used in the field (for instance RNA-Site [1]).
> - We discard binding sites for multiple ligands to keep the RNA-Ligand task multi-class and not multi-label.
> - You are entirely right that bi-modal is not the correct term, but rather bi-group, with groups defined as you mentioned.
> - Unobserved pairs are assumed to be negative (which is indeed partly erroneous), as is common in the field (SMRTBind, RNAmigos2, GeRNAbind..)
> - The RNA-VS task's objective is to achieve similar results to docking in a much faster way (miliseconds instead of minutes). We will mention the limited performance of rDock and synthetic data as a limitation
> - The train/validation/test split ratios are fixed to be around .7/.15/.15 except for existing datasets where we use the existing splits
> - The gRNAde dataset does not remove redundancy and treats multi-chain RNAs individually, which explains why it holds over 11k RNAs
> - The RNA-Prot task is useful as a benchmark task. It could however be useful for peptide design (like small molecule binding site prediction) and symetric versions exist on proteins (RBP prediction).
> - "1D sequence models, such as LSTM or Transformers, use only the raw nucleotide sequence.", We will add an explicit mention of one hot encoding for our models.
> - In Figure 2, the GraphRepresentation corresponds to graph representations (by default 2.5D which is top performing).
> - Number of parameters and stopping criterions will be included in later versions.
> - All models' hyperparameters were reported in Appendix F, tables 4,5,6
> - CD-HIT and US-align are called by python objects available in the code (default parameters) which can be tuned if needed.
> - The “recomputation” option allows to build the benchmark datasets from the PDB which means that the datasets can be automatically updated with new PDB entries. To ensure reproducibility, he official versions are built from fixed data releases.
> - Our current experiments unambiguously show that for a fixed amount of data, sequence is outperformed by structure-based models. We acknowledge that this result is quite expected, but also propose other actionable conclusions, notably on the performance of 2D vs 3D representations, or with regards to the relevance of the non-canonical interactions for RNA structure modeling (2.5D vs 2D). We believe these results to be very relevant for the field.
> - We ran experiments on the use of pre-trained sequence embeddings (RNA-FM) either in sequence or structure-based models. However, we did not include those results because we could not find a positive impact on performance, to our strong surprise. We believe this behavior necessitates a dedicated study, orthogonal to the point of this paper: introducing datasets & benchmarks.
>
> We hope to have answered all questions, and will do our best to include this feedback as well as all "minor" points in later versions of our work.
> We thank you again and remain at your disposal should you have more questions.
>
> ---
> [1]Hong Su, Zhenling Peng, and Jianyi Yang. Recognition of small molecule–rna binding sites using rna sequence and structure. Bioinformatics, 37(1):36–42, 2021.

---

> > ### Comment · Reviewer_Q7gt · 2025-11-23
> > **Response to authors**
> >
> > Thank you for the detailed responses, I appreciate the clarifications provided. I agree that the primary area for this work is indeed “Datasets and Benchmarks.” My remaining concern is about the possibility that a highly simplified benchmark may unintentionally encourage score-driven ML work without sufficient biological understanding. Because of this, it becomes especially important to make the rationale behind each design choice very clear and to explicitly highlight any limitations or simplifications that users should be aware of.
> >
> > Before going into the specific points, I also want to note that since the manuscript has not yet been updated, it is somewhat difficult to follow the clarifications in isolation. Incorporating these explanations directly into the paper (many of which appear to be small textual updates) would make it much easier to understand how the final version of the benchmark is intended to work.
> >
> > Below, I list the follow-up points where I still feel some uncertainty or where the motivation was not entirely clear from either the paper or the rebuttal. For the items where your answers fully resolved the issue, I will skip them here. I would also appreciate responses to the questions raised in the “Questions” section of the original review.
> >
> > 1. It would be helpful to clarify the basis for defining RNA fragments as <15 nt. Is this a convention established in another dataset or benchmark? Likewise, I still do not fully understand the rationale for excluding RNAs that “depend on protein interactions,” or how this dependence is identified. Some RNAs adopt stable folds only in the presence of proteins, so I am trying to understand the intended biological assumption here.
> > 2. For the distance cutoff: if 8 Å is standard for defining binding sites in the literature (as used in RNA-Site), could you comment on why 4 Å was used earlier in the workflow (line 119)? It would help readers if the reasoning behind using different cutoffs at different steps were made explicit.
> > 3. Regarding the removal of binding sites that bind multiple ligands: I understand the desire to keep the task multi-class rather than multi-label, but this seems to simplify the biology in ways that could matter for downstream use cases. Many RNA pockets do engage with chemically distinct ligands, and excluding such cases removes real examples rather than natural ambiguity. If this design choice is motivated by modeling convenience rather than biological reasoning, it would be valuable to acknowledge it explicitly so users understand what class of true interactions is being filtered out.
> > 5. On negative labels: if unobserved RNA-ligand pairs are treated as negatives, I think this assumption should be stated explicitly in the paper and in the framework documentation, as it introduces a specific type of label noise. Since you mention that this follows conventions used in SMRTBind, RNAmigos2, and GeRNAbind, including these citations in the main text would help situate the decision and make clear that the benchmark inherits a common (but imperfect) assumption from prior work.
> > 9. I am still unsure about the exact input to the RNA-Prot task. Does the model receive only the RNA (sequence or structure), with no representation of the interacting protein? If so, a brief explanation of the intended rationale would help significantly. Intuitively, the protein partner seems crucial for determining whether a nucleotide is likely to be in contact, so I am trying to understand what signal the model is expected to learn from RNA alone, and how users should interpret predictions in this setup.
> > 16. Regarding the baselines: Table 4 and Table 5 list several hyperparameters, but they do not provide total parameter counts or the training-termination criteria. Without this information, it is difficult to determine whether performance differences reflect representation choices or simply differences in model capacity or training duration. Even approximate parameter counts and a note on whether training used fixed epochs or any form of early stopping would help make the baselines more interpretable and reproducible.
> > 17. The clarification about RNA-FM is useful, and I agree that understanding why sequence-pretrained embeddings do not help could merit a separate study. At the same time, I wanted to note that RNA-FM is one of several available RNA LMs, and several more recent models have shown stronger downstream performance. Because the field of RNA pretraining is evolving quickly, even a brief mention that multiple pretrained models were considered, but only RNA-FM was evaluated, would help avoid readers concluding that sequence pretraining is broadly ineffective for these tasks.
> >
> > I hope these points are helpful as you refine the benchmark. I appreciate the effort put into the rebuttal and the willingness to engage with these details.

---

### Official Review · Reviewer_4eHp · 2025-10-31

**Soundness:** 2
**Presentation:** 3
**Contribution:** 2
**Rating:** 2
**Confidence:** 5

**Summary:**

This paper introduces a standardized and reproducible benchmark suite for RNA 3D structure–function modeling. It provides seven tasks, modular data-processing tools, and baseline results across multiple RNA representations using experimental RNA structures. The work is aiming to facilitate future model development and fair comparison in the emerging field of RNA structure-function prediction.

**Strengths:**

•	The paper addresses an important and underexplored problem: the lack of standardized and reproducible RNA 3D structure–function benchmarks. While protein benchmarks have driven major progress in deep learning for structural biology, RNA has remained comparatively neglected. Establishing such a benchmark is useful for advancing model development in this area.
•	The authors provide open-sourced code, data, and results (via an anonymous repository) and report results over multiple random seeds with error bars, which supports transparency and reproducibility.
•	The paper is well-written, and easy to follow.

**Weaknesses:**

1.	Limited technical novelty: The main distinction claimed over prior works (e.g., Beyond Sequence, Xu & Moskalev et al., ICLR 2025; rnaglib, Mallet et al., 2021) is the use of experimental RNA data rather than predicted 3D RNA structures. While this is useful, it represents a relatively modest incremental step rather than a substantial methodological or conceptual advance.
2.	Substantial overlap with existing benchmarks and findings: The paper’s key conclusions largely align with prior work and do not provide new insights:
o	The finding that atomic-level detail may be unnecessary for modeling RNA structure under limited data is already reported in Beyond Sequence (Xu & Moskalev et al., ICLR 2025).
o	The claim that 3D methods outperform 1D methods contradicts the authors’ assertion of novelty, since Beyond Sequence already demonstrated that 3D models outperform 1D models when sufficiently parameterized and with sufficient receptive field (see Table 1 in their paper with nucleotide pooled 3D models which outperform 1D models almost always) and when reliable structures are available. Also this conclusion is rather expected. With reliable 3D structures, it is not a surprise that 3D models work better than 1D models as they have all the information that 1D models have and more.
o	The observation that 2.5D representations perform best is consistent with previous findings as well (e.g., Beyond Sequence’s Transformer 1D2D model also outperformed most of the 1D and even 2D and 3D models at times).
Thus, overall, the empirical conclusions reinforce already well-reported conclusions rather than providing new insights.
3.	Unclear distinction from rnaglib: The paper builds directly on rnaglib (Mallet et al., 2021), which already introduced a Pythonic benchmarking framework for RNA-related tasks with modular dataset, splitting and evaluation components. The current work seems to primarily use rnaglib and introduces additional datasets within this framework rather than introducing fundamentally new functionality or benchmarking paradigms.
4.	Limited scope of evaluated models: The benchmark includes only a small set of baseline models — LSTM and Transformer for 1D, RGCN and GVP(-2.5D) for 2D and 3D — which makes the comparative conclusions less robust. In contrast, recent works such as Beyond Sequence (Xu & Moskalev et al., ICLR 2025) evaluated a substantially broader set (multiple 1D, 2D, and 3D architectures, including both spectral and spatial GNNs for 2D and classical as well as quite recent 3D models including GVP). Without testing diverse 2D and 3D models, it is difficult to conclude that 2.5D representations universally outperform 3D ones. Also, given that the curated data is from RCSB-PDB database which is known to have structures of different structural resolutions, it is possible that there are still structural inaccuracies in the annotated structures and hence different 2D models may still work better than 3D models as reported in prior works.
5.	Practical relevance and generalizability: The paper acknowledges that high-quality RNA structures are scarce, yet focuses solely on curated experimental structures (which are still a few thousand). It is unclear how conclusions drawn from such high-quality but limited data translate to real-world scenarios, where RNA data are often noisy and with incomplete labels. Prior works such as Beyond Sequence (Xu & Moskalev et al., ICLR 2025) explored this robustness dimension (e.g., noisy structures, sequencing errors), which this paper omits entirely.

**Questions:**

See weaknesses above where I have described my rational for questions as well. Specifically, the following questions are raised by the weaknesses described above.
1.	Clarify the novelty over rnaglib: What specific features or design principles differentiate this benchmark suite from rnaglib beyond the addition of datasets?
2.	Positioning with respect to Beyond Sequence: The paper’s results and claims of novelty should be clearly delineated from Beyond Sequence (ICLR 2025). How does this work contribute new insight or capability beyond confirming previously established trends beyond just using experimental structures?
3.	Broader model coverage. Comparing against more recent 2D and 3D architectures (e.g., E(n)-GNN, EGNN, SE(3)-Transformer, or equivariant message-passing baselines) will provide a more comprehensive benchmark comparison. Similarly, comparisons against recent 2D, 2.5D and 1D models should also be made.
4.	Real-world robustness: Testing model robustness to predicted or noisy RNA structures (e.g., from RoseTTAFoldNA, EternaFold, or low-resolution PDB entries) would make the benchmark more reflective of practical challenges.
Some additional questions that are necessary for the benchmark to be comprehensive and useful from bioinformatics applications perspective are as follows. It would be good to clarify them as well.
5.	Given limited availability of high-resolution RNA structures, how do you ensure diversity across RNA families? How sensistive are results to incomplete or noisy annotations from Rfam?
6.	Given that the claim is that the datasets used in the study are high quality experimental structures, then why might 3D models underperform compared to 2.5D ones?
7.	How do the  RNA-ligand interactions tasks (RNA-Ligand, RNA-Prot, RNA-site) account for structural flexibility?
8.	Unless I am mistaken, the dataset appears to be mostly dominated by tRNA and rRNA families. Will this limit the benchmark’s ability to generalize to diverse RNA types.
Given the many weaknesses which need thorough addressing and many more experiments to be conducted, I feel this paper is not ready yet for publication but can benefit from iterations for future conferences.

---

> ### Author Response · Authors · 2025-11-17
>
> We thank the reviewer very much his detailed analysis of our work. The reviewer raises some important concerns that we hope to mitigate by answering the questions provided.
>
> __Q1: Difference from RNAglib__
>
> Our work significantly differs from rnaglib in multiple ways, that are extensively discussed in Appendix A. Notably, rnaglib did not include modular datasets, splitting, nor evaluation components. The confusion might stem from the integration of our new benchmarking tool within the RNAglib library. While it is tempting from a marketig perspective to develop a new "brand" for each piece of work, we felt that the community is better served by expanding on existing resources, thereby ensuring interoperability, maintenance, and ease-of-use.
>
> Shortly put, while rnaglib mostly provided a datastructure for RNA structural data, this work provides an extensive toolkit for benchmarking ML models on RNA structures (including data splitting and preprocessing tools, reproducible tasks, well-documented modular software allowing the fast creation of new tasks, and more).
>
> __Q2: Difference from "Beyond Sequence"__
>
> While we agree this paper serves an aligned goal, the use of experimental data is a crucial difference. Since PDB data is famously hard to parse, making it accessible to users is an important contribution. This is all the more important in a setting where structure prediction methods still mostly fail on RNA [1], especially when used without MSA as in paper [1].
>
> More importantly, [1] did not make their dataset public, effectively **making our work the only publicly available benchmark on RNA structure**. Our datasets, contrary to [1], are available on Zenodo, our code is open-source and modular, allowing easy extension to other tasks, and benchmarking of new models.
>
> The reviewer is right in pointing out overlap in the insights arising from this work and "Beyond structure". However, those results are obtained on an independent dataset and set of tasks, and therefore consolidate their findings. Moreover, part of their conclusions were mitigated by the potential impact of weak predicted 3D structure, an issue settled by our results.
>
> Actually, our conclusions are more subtle than a pure "2D vs 3D" results. In our experiments, the simple 2D representation proposed in their paper actually **performs on par with 3D** structure (probably due to the poor performance of structure prediction methods).
>
> Going beyond the simple unoriented graphs, including backbone direction and edge types (2D+) resulted in a strong, consistent improvement. In addition to that, we report the **first** systematic assesment of the 2.5D edges, resulting in the strongest performance and suggesting a promising way forward for RNA structure modeling.
>
> Those results are a strict expansion over prior work.
>
> __Q3: More models__
>
> This is a valid point. Plese note however that we chose to focus our attention on extensively exploring hyperparameters for the models discussed. We will include more benchmarks in future iterations of this work.
>
> __Q4: Robustness to low resolution__
>
> Splitting our performance on resolution is a very good idea. Thank you for bringing it to our attention. The assessment of model performance on predicted structure should use the "Beyond sequence" benchmark, if ever made public.
>
> __Q5/8: Family bias__
>
> We provide detailed information on family distributions of our tasks in Figure 9 of the appendix. While multiple RNA families are represented, we are not able to completely correct for the biased distribution of the PDB. This is a limitation of our tool, however one that is inherent to the field.
>
> Could you clarify your point on sensitivity with regards to incomplete Rfam annotations? To our understanding, Rfam annotations are only relevant to the RNAFamily task, where unannotated RNAs are removed from the dataset.
>
> __Q6: Performance of coarse-grain vs full atom__
>
> We propose two hypotheses for the performance of 2.5D approaches in the "Model Benchmarking" section.
> From a biological point of view, we hypothesize that the flexible nature of RNA structures may be better reflected in a graph representation compared to using atomic coordinates, which only ever are a "snapshot" of an RNA in a particular, often interaction-mediated, conformation.
>
> From a computational point of view, graphs are an efficient data structure with a sparsity prior, which might be especially relevant for use cases with limited limited data.
>
> However, further research is needed to fully elucidate this question.

---

> ### Author Response · Authors · 2025-11-17
>
> __Q7: RNA Structural flexibility__
>
> Structural flexibility is not explicitly considererd by our framework. This is a challenge that is plaguing the entire field of ML for structural biology, since the data format (and experimental pipelines) currently used (PDB/mmCIF) treats biological macromolecules as static. pLDDt and other proxies for flexibility are still relatively poorly understood, especially for RNA. Yet, we fully agree that better accounting for structural flexibility will be a major leap for the field in the future. As a first step, we added to our framework the possibility to implement the multi-graph representation proposed in gRNAde [3].
>
> It is expected that model performance will be poorer on RNA families that are rare in the training data. Unfortunately, we are unsure to which extent this problem can be mitigated in the absence of new data.
>
>
> ### Summary
>
> We believe the main points of the reviewer rooted from a misunderstanding between our contribution and existing work (Q1/2) which we will emphasisze in future versions. Other points include biological considerations inherent to the field (Q5,7,8), including more models and performance analysis split by resolution, as well as a question on the performance of coarse grained methods.
>
> Again, we would like to thank for the constructive feedback.
> Our work does represent the first publicly available dataset on RNA structures. We also propose novel insights on RNA structure representations, showing the advantage of the use of the 2D+ or 2.5D representations.
>
> We hope that we were able to clarify some aspects of our work and remain at your disposal for further questions.
>
> ---
>
> [1] Xu, Junjie, et al. "Beyond sequence: Impact of geometric context for rna property prediction." ICLR (2025)
>
> [2] Assessment of nucleic acid structure prediction in CASP16, Kretsh et al. biorxiv, 2025
>
> [3] gRNAde: Geometric Deep Learning for 3D RNA inverse design, Joshi et al. ICLR (2025)

---

### Official Review · Reviewer_1tuq · 2025-11-01

**Soundness:** 2
**Presentation:** 2
**Contribution:** 3
**Rating:** 2
**Confidence:** 3

**Summary:**

This paper introduces a benchmarking suite of seven tasks for evaluating deep learning models on RNA 3D structure-function prediction, built on the rnaglib Python package with modular data processing, rigorous splitting strategies, and standardized evaluation protocols.

**Strengths:**

Addresses the lack of standardized benchmarks for RNA 3D structure–function research

Provides seven benchmark tasks covering distinct biological challenges, along with modular tooling (filters, splitters) to facilitate research

Emphasizes strict data-splitting strategies (sequence- and structure-similarity–based) to prevent leakage, and empirically demonstrates performance inflation under random splits

**Weaknesses:**

The paper acknowledges that its simple baseline model (RGCN) does not reach SOTA performance compared with models in the literature

Some tasks (e.g., RNA-CM) have very small datasets, which may limit the effectiveness of training more complex models.

The datasets mirror the overrepresentation of tRNAs and rRNAs in the PDB, potentially biasing models toward these families.

**Questions:**

Why does 2.5D beat 3D? The finding that coarse-grained graphs outperform 3D atomic graphs raises the question: is this due to intrinsic properties of RNA data, or limitations in the current 3D modeling approaches?

What performance is “good enough”? For application-driven tasks like RNA-VS (virtual screening) , is the current baseline performance (e.g., AUROC 0.759) sufficient to reliably guide real-world drug discovery workflows?

---

> ### Author Response · Authors · 2025-11-17
>
> We would like to thank the reviewer for their critical assessment of our work. It raises some important limitations, but, in our opinion, does not do justice to our intention of providing a benchmarking suite. We provide a short explanation to each point below.
>
> __W1:__ Indeed. As providers of a benchmarking suite our focus did not lay on achieving SOTA results but on providing robust and reproducible tasks. Having submitted in the "datasets & benchmarks" category, we feel our contribution should be evaluated on the usefulness and useability of the tasks we provided, not the performance of our example models.
>
> __W2:__ This is correct. The data availability of RNA 3D structures so far prevents the training of large, complex models. Still, even in a regime of limited data availability, it is of interest to assess what is possible now, and build the infrastructure that can accommodate new structural data as it is generated and/or predicted.
>
> __W3:__ Again, this is also correct. This bias reflects the current state of knowledge in RNA structural biology and is thus, in our opinion, not a weakness specific to our work. Moreover, our tool easily allows the creation of new datasets excluding or undersampling tRNAs and rRNAs, should that be desired.
>
> __Q1:__ We propose two hypotheses for the performance of 2.5D approaches in the "Model Benchmarking" section: "We hypothesize that atomic-level detail is unnecessary for modeling inherently unstable RNA structures, particularly with limited data. This is in line with existing findings on predicted structure [1] [...] Moreover, our results demonstrate that base-pairing information is a crucial prior for both 2D and 3D models, consistent with previous results obtained on a virtual screening task on graph representations. The 2.5D representation was the top performer, likely due to its combination of a powerful biological prior (non-canonical pairs) with an efficient, sparse graph structure.
>
> However, we unfortunately have no way to measure whether the benefits stem primarily from the biological or the methodological prior.
>
> __Q2:__ This is an important point. Indeed, the current baseline performance is insufficient for direct application in real-world drug discovery workflows. However, the first step in building models with real world applications lies in constructing meaningful baselines. Something we hope to have achieved.
>
>
> ### Summary
>
> The reviewer agrees there is a "lack of standardized benchmarks for RNA 3D structure–function research". Our paper proposes a benchmark suite to fill in that gap, in line with ICLR scope that includes "datasets & benchmarks".
>
> We believe the weaknesses pointed out are inherent to the field of RNA structural biology, such as limited data (W2,W3), or to the benchmark nature of our contribution (W1,Q2).
>
> We hope to have answered your questions and remain at your disposal for any follow-up.
>
> ---
> [1] Xu, Junjie, et al. "Beyond sequence: Impact of geometric context for rna property prediction." ICLR (2025)

---

### Official Review · Reviewer_hCnj · 2025-11-03

**Soundness:** 2
**Presentation:** 3
**Contribution:** 2
**Rating:** 4
**Confidence:** 3

**Summary:**

The paper introduces a modular and comprehensive benchmark for RNA 3D structure–function modeling, extending the rnaglib framework. It defines seven standardized tasks covering biological function prediction, molecular design, and RNA–ligand interaction modeling. Each task includes curated datasets, redundancy filtering, similarity-based data splitting, and consistent evaluation metrics. The authors provide a unified interface for data loading, preprocessing, and model benchmarking, together with baseline results. All datasets, code, and documentation are openly available.

**Strengths:**

The work fills a clear gap in the current landscape of molecular machine learning benchmarks by focusing on RNA 3D structures. The design is comprehensive and reproducible, addressing crucial challenges such as data leakage, redundancy, and the lack of standardized evaluation protocols.

The dataset construction is biologically meaningful and diverse, spanning multiple functional levels from sequence to small-molecule binding. The implementation is accessible and well-documented, offering an important resource for the community. The baseline experiments are thorough and demonstrate that the proposed tasks are feasible and sufficiently challenging.

**Weaknesses:**

Some details are missing for the curated datasets. The authors should clearly present detailed statistics of the datasets for each task in the main text. For example, the size of the positive and negative samples in RNA-VS is currently unknown, which is important for understanding the data balance.

Also, the split strategies could be described in more detail. For instance, the RNA-LIGAND dataset is said to be split based on structural similarity, but the exact similarity threshold is not stated.

The related work section also misses structure-based RNA models [1] [2].

[1] Stefaniak, Filip, and Janusz M. Bujnicki. AnnapuRNA: A scoring function for predicting RNA-small molecule binding poses. PLoS computational biology, 2021

[2] Shuo Zhang, Yang Liu, and Lei Xie. Physics-aware graph neural network for accurate rna 3d structure prediction. NeurIPS 2022 Workshop on Machine Learning for Structural Biology, 2022

**Questions:**

Are the datasets used in LigandRNA and AnnapuRNA possibly included or overlapped with the proposed benchmark? Both datasets are designed for evaluating or scoring RNA–small molecule binding poses.

---

> ### Author Response · Authors · 2025-11-17
>
> We sincerely thank the reviewer for their thoughtful comments and for acknowledging the fact that our work covers a real need in the field, while providing tasks that are not only suitable to the ML practitioner but also have real biological relevance.
>
> Thank you for bringing to our attention the missing class distribution and splitting threshold. We will include these numbers in a future version of the work (The similarity threshold for RNA-LIGAND is set at a USAlign score of 0.5).
>
> AnnapuRNA and LigandRNA are important works in RNA-ligand pose prediction. As of now, ligand pose prediction is not included in the seven proposed tasks. However, this represents a very interesting task to include in the future.
>
> Given the overall positive appreciation of our work, and the very minor weaknesses pointed out, we were surprised by the recommendation of the reviewer. We are at your disposal to discuss any concerns you may have.

---

### Meta-Review · Area_Chair_4fvf · 2026-01-05

**Summary:**

The reviewers consistently cited a lack of technical novelty and significance for a machine learning venue as the primary grounds for rejection, characterizing the work as an incremental infrastructural extension of existing frameworks that overlaps with concurrent findings rather than offering distinct methodological advances. Substantial concerns were raised regarding the robustness and real-world applicability of the datasets, specifically highlighting their small size, heavy biological bias toward tRNA and rRNA families, and the exclusive reliance on experimental structures without addressing the critical challenge of modeling noisy or predicted data. Furthermore, the evaluation framework was criticized for utilizing a limited set of baseline models that fail to capture the current state-of-the-art, while the manuscript was noted to lack essential details regarding dataset statistics, splitting protocols, and preprocessing rationale.

**Reviewer Concerns:**

The authors did not adequately resolve critiques about the exclusive reliance on small, heavily biased experimental datasets, which reviewers felt limited the benchmark's utility for real-world scenarios where predicted structures are common. The refusal to expand the baseline evaluations to include state-of-the-art architectures left the limited scope concern active, and specific biological justifications remained unconvincing to the expert reviewer.

**Reviewer Scores:**

The final score would solidify at 2, 2, 2, 4. Reviewer hCnj would likely maintain score of 4, as the authors effectively addressed their specific requests for dataset statistics and split thresholds. Conversely, the remaining three reviewers would resolutely adhere to their scores of 2: Reviewer 1tuq would view the authors' admission that dataset biases are inherent to the field as a confirmation of the benchmark's limited utility; Reviewer 4eHp would remain unpersuaded regarding the technical novelty, viewing the work as a minor incremental update to the rnaglib library rather than a substantive ICLR contribution; and Reviewer Q7gt, despite acknowledging clarifications, explicitly retained significant reservations about the benchmark's biological simplifications (such as excluding protein contexts) that undermine its real-world relevance.

---

### Decision · Program_Chairs · 2026-01-26

Reject